# Cdc48-like protein of actinobacteria (Cpa) is a novel proteasome interactor in mycobacteria and related organisms

Michal Ziemski[1], Ahmad Jomaa[1], Daniel Mayer[2], Sonja Rutz[1], Christoph Giese[1], Dmitry Veprintsev[2†], Eilika Weber-Ban[1*]

[1]Institute of Molecular Biology & Biophysics, ETH Zurich, Zurich, Switzerland; [2]Laboratory of Biomolecular Research, Paul Scherrer Institute, ETH Zurich, Villigen, Switzerland

**Abstract** Cdc48 is a AAA+ ATPase that plays an essential role for many cellular processes in eukaryotic cells. An archaeal homologue of this highly conserved enzyme was shown to directly interact with the 20S proteasome. Here, we analyze the occurrence and phylogeny of a Cdc48 homologue in *Actinobacteria* and assess its cellular function and possible interaction with the bacterial proteasome. Our data demonstrate that Cdc48-like protein of actinobacteria (Cpa) forms hexameric rings and that the oligomeric state correlates directly with the ATPase activity. Furthermore, we show that the assembled Cpa rings can physically interact with the 20S core particle. Comparison of the *Mycobacterium smegmatis* wild-type with a *cpa* knockout strain under carbon starvation uncovers significant changes in the levels of around 500 proteins. Pathway mapping of the observed pattern of changes identifies ribosomal proteins as a particular hotspot, pointing amongst others toward a role of Cpa in ribosome adaptation during starvation.
DOI: https://doi.org/10.7554/eLife.34055.001

*For correspondence: eilika@mol.biol.ethz.ch

Present address: †Centre of Membrane Proteins and Receptors, University of Birmingham and University of Nottingham, Nottingham, United Kingdom

Competing interests: The authors declare that no competing interests exist.

## Introduction

Energy-dependent chaperones and chaperone-protease complexes comprise important cellular components guarding protein homeostasis in all kingdoms of life. Chaperones in the context of protein degradation form complexes with compartmentalizing protease cylinders and act by unfolding their protein substrates, translocating them into the proteolytic compartment for degradation. Mycobacteria and many other actinobacteria possess several independent chaperone-protease complexes: the canonical bacterial Clp protease system and membrane-bound FtsH protease as well as a eukaryotic-like proteasome system (*Imkamp et al., 2015*). The existence of proteasomes in bacteria is an unusual feature restricted to actinobacteria, a diverse phylum with different members showing various examples of eukaryotic-like complexes or activities (*Cavalier-Smith, 2002*). Like in eukaryotes, the bacterial 20S proteasome core particle must associate with an activator to form the fully active proteolytic complex. The first activator described was ARC (ATPase forming ring-shaped complexes), also called Mpa (mycobacterial proteasomal ATPase) in mycobacteria, a divergent AAA+ family protein that catalyzes the ATP-dependent unfolding and translocation into the proteasome chamber of proteins post-translationally modified with prokaryotic ubiquitin-like protein Pup (*Burns et al., 2009*; *Pearce et al., 2008*; *Striebel et al., 2014*). Proteomic studies on whole-cell purified 'pupylomes' (pupylated proteomes) identified hundreds of cellular proteins as pupylation targets (*Compton et al., 2015*; *Festa et al., 2010*; *Watrous et al., 2010*). One of the best characterized pupylation targets and favorite in vitro substrates is ketopantoate hydroxymethyltransferase (PanB), the enzyme that catalyzes the first committed step of pantothenate biosynthesis (*Chaudhuri et al., 2003*; *Pearce et al., 2006*).

**eLife digest** Cells use proteins to carry out the biological processes necessary for life. If a protein becomes damaged or is no longer needed, cells must dispose of it, just as we might take out the trash. The cell's main 'garbage disposal unit' is the proteasome, a barrel-shaped molecular machine that breaks down unwanted proteins. The proteasome binds to other molecules called regulators, which select the proteins to be dismantled.

The proteasomes of mycobacteria – a group that includes the bacteria that cause tuberculosis – help them to survive hostile or rapidly changing environments. Mycobacteria contain a molecule called Cpa whose structure is like a regulator that is found in many non-bacterial cells. Ziemski et al. therefore set out to investigate whether Cpa performs a similar role in bacteria.

The results of biochemical experiments performed in test tubes revealed that Cpa forms rings made up of six copies of itself. These rings can bind to proteasomes. Ziemski et al. also created genetically modified mycobacteria that could not produce Cpa and studied how they coped with starvation. These modified bacteria stopped growing sooner than their similarly starved genetically normal counterparts. The two groups of bacteria also produced different amounts of some proteins.

Ziemski et al. used a technique that pulled Cpa out of the starving genetically normal cells to analyse the proteins that Cpa physically interacts with. These proteins included building blocks of the ribosome, the cellular machinery that produces new proteins. It therefore appears that Cpa helps mycobacteria to cope with starvation by reducing the amount of protein made by the cell.

Cpa may also help mycobacteria to survive in other stressful conditions, such as those that the bacteria experience when they infect the human body. Developing drugs that prevent Cpa from working could therefore potentially lead to new treatments for a number of diseases caused by mycobacteria, such as tuberculosis.

DOI: https://doi.org/10.7554/eLife.34055.002

More recently, a second proteasomal activator was discovered, the bacterial proteasome activator Bpa (also referred to as PafE) (*Delley et al., 2014*; *Jastrab et al., 2015*). Bpa acts in an ATP-independent manner and is thought to aid in proteasomal degradation of proteins damaged under stress conditions, since it supports degradation of the unstructured model substrate casein and is involved in degradation of the conformationally unstable transcriptional heat shock protein repressor HspR in vivo (*Jastrab et al., 2017*; *Jastrab et al., 2015*). The two non-homologous activators Mpa/ARC and Bpa both form ring-shaped complexes with a central pore and share as one additional important feature with eukaryotic proteasome interactors a C-terminal proteasome interaction motif featuring a penultimate tyrosine (*Delley et al., 2014*; *Imkamp et al., 2015*). In eukaryotes, this motif is called HbYX motif, since the tyrosine is preceded by a hydrophobic residue. Mpa/ARC and Bpa on the other hand both carry the sequence 'GQYL', where Q can be mutated freely without changing association behavior of the activator with the proteasome (*Jastrab et al., 2015*).

In eukaryotes, 20S proteasomes are found in the cytoplasm as well as in the nucleus, where they degrade proteins upon association with the 19S regulatory particle. The regulatory particle contains six ATPase subunits that form the hexameric basal ring stacking onto the 20S cylinder. However, the proteasome is also responsible for degradation of ER-resident proteins via the so-called ER-associated degradation (ERAD) pathway that requires retrotranslocation of substrate proteins out of the ER into the cytoplasm. In this context, the 20S proteasome additionally cooperates with another AAA+ protein, Cdc48 (also known as p97 or VCP) (*Baek et al., 2013*; *DeLaBarre et al., 2006*; *Wolf and Stolz, 2012*). Cdc48 has also been implicated in a multitude of other cellular processes like membrane fusion, autophagy and gene expression (*Baek et al., 2013*; *Yamanaka et al., 2012*). The different Cdc48 functions are largely mediated by various adaptor proteins (>40) binding to the N-terminal domain or C-terminal region of Cdc48 and aiding in recognition of different substrates (*Baek et al., 2013*; *Buchberger et al., 2015*). Although the exact role played by Cdc48 during ERAD is not fully understood, it is involved in pulling proteins from the membrane or from ribosomes during co-translational degradation, and recent evidence suggests even a direct association with the 20S particle (*Barthelme and Sauer, 2013*). Such a direct role as 20S proteasome activator has been shown for the Cdc48 homolog from archaea in vitro (*Barthelme et al., 2014*). Archaeal Cdc48 forms

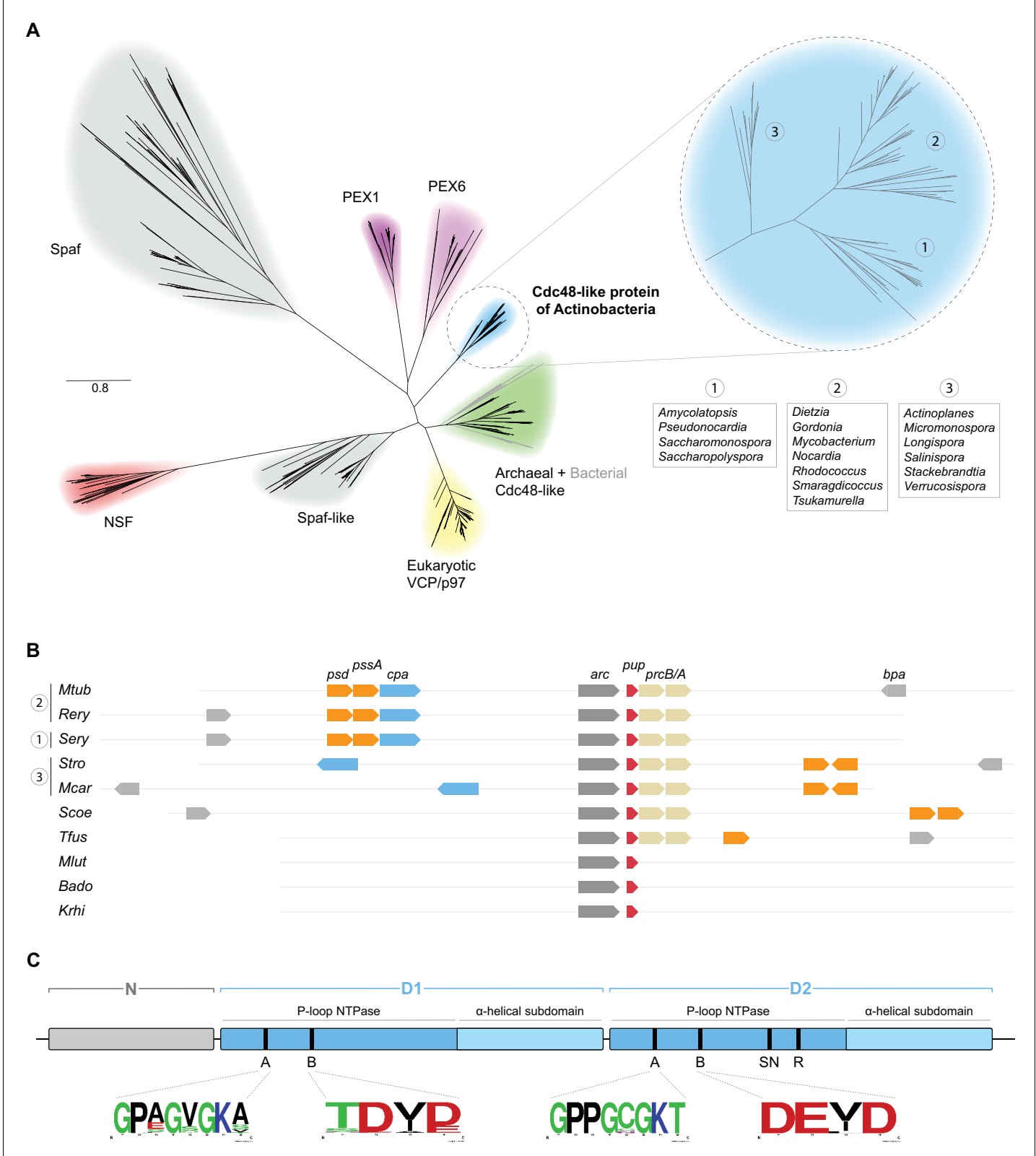

**Figure 1.** Bioinformatical analysis of actinobacterial Cpa. (**A**) Phylogenetic relationships of different members of the Cdc48 protein family to the novel actinobacterial homolog. The peroxisome biogenesis factor 1 and 6 families (PEX1 and PEX6) are depicted in magenta, spermatogenesis-associated factor (Spaf) and Spaf-like families are colored grey, the archaeal Cdc48 homolog family is shown in green and N-ethylmaleimide-sensitive fusion protein (NSF) family in red. The Cdc48-like protein of actinobacteria (Cpa) forms a separate tight cluster (blue), while sporadically occurring homologs in

*Figure 1 continued on next page*

*Figure 1 continued*

other bacteria fall within the archaeal cluster (grey branches within the green cluster). (**B**) Occurrence and arrangement of the *cpa* gene locus in selected actinobacterial genomes. (**C**) Domain arrangment of Cdc48-like protein. The protein features an N-terminal domain (N-domain, grey) followed by two consecutive AAA modules (D1 and D2, blue). Each of the modules can be further subdivided into a P-loop NTPase domain and an α-helical subdomain. Conserved motifs are abbreviated as: A – Walker A motif, B – Walker B motif, SN – sensor asparagine, R – arginine finger. Organism names are abbreviated as follows: *Mtub* – Mycobacterium tuberculosis, *Rery* – Rhodococcus erythropolis, *Sery* – Saccharopolyspora erythraea, *Stro* – Salinispora tropica, *Mcar* – Micromonospora carbonacea, *Scoe* – Streptomyces coelicolor, *Tfus* – Thermobifida fusca, *Mlut* – Micrococcus luteus, *Bado* – Bifidobacterium adolescentis, *Krhi* – Kocuria rhizophila.

DOI: https://doi.org/10.7554/eLife.34055.003

The following figure supplement is available for figure 1:

**Figure supplement 1.** Principal component analysis of Cdc48 family protein sequences.

DOI: https://doi.org/10.7554/eLife.34055.004

a complex with the 20S proteasome that is capable of degrading ssrA-tagged model substrates, likely due to the disordered, extended nature of the ssrA-tag at their C-terminus (*Barthelme and Sauer, 2013*). Eukaryotic Cdc48 on the other hand is not able to recognize ssrA-tagged substrates in vitro, which was proposed to be due to the absence of two hydrophobic residues in the entry pore loop. These residues must be involved in recognition of the ssrA tag, since their introduction into the eukaryotic p97 enabled it to unfold ssrA-tagged model substrates (*Barthelme and Sauer, 2013*; *Rothballer et al., 2007*). Although the ssrA-tag is not a bona fide degradation signal in eukaryots and archaea, the D1 pore-1 loops and the nature of the residues within these loops appear to play a general role in substrate recruitment. A recent study demonstrated that mutating residues in these entry loops in vivo deregulated cellular protein turnover (*Esaki et al., 2017*). In the mammalian homolog, the same residues are required for proper function of the ERAD pathway (*DeLaBarre et al., 2006*).

In this study, we characterize the Cdc48-like protein of actinobacteria (Cpa) by a combination of bioinformatical analysis, genetic modification and biochemical characterization to explore ring formation of Cpa, to probe its ability to interact with the 20S proteasome as well as to map proteomic changes resulting from its deletion in *Mycobacterium smegmatis* (Msm). We show that Cpa forms hexameric rings, their assembly strongly dependent on pH and ionic strength. Once formed, the rings are capable of interacting with the proteasomal core particle, identifying Cpa as a novel proteasomal interactor in actinobacteria. Moreover, we report the comparative proteomic profile of an Msm parent and *cdc48*-deficient strain, providing a basis for assessing the function of this AAA + ATPase in mycobacteria.

## Results

### Phylogeny of Cdc48-like protein from actinobacteria (Cpa)

Actinobacteria encode a number of AAA+ proteins that were shown to be involved in protein degradation pathways, amongst them the chaperone complexes of the Clp protease system, ClpX and ClpC, as well as the energy-dependent proteasome activator Mpa in mycobacteria (*Laederach et al., 2014*). As Cdc48 in eukaryotes and archaea is amongst other functions also tied to protein degradation, we set out to investigate if its homolog might have an analogous function in actinobacteria. The *Mycobacterium tuberculosis* (Mtb) Cpa protein Rv0435c is annotated in the UniProt database as Cdc48 based on sequence homology to the eukaryotic Cdc48 (p97 or VCP) (*The UniProt Consortium, 2017*). In order to gain a better understanding of the position of the actinobacterial homolog within the family of Cdc48-like proteins we performed a multiple sequence alignment of 1167 sequences including members of five previously identified eukaryotic Cdc48 families (PEX1/6, NSF, Spaf, Spaf-like, p97) as well as archaeal and actinobacterial Cdc48-like proteins using the ClustalO algorithm. The alignment was used for construction of a phylogenetic tree using the approximately-maximum-likelihood method of the FastTree software (*Price et al., 2010*). As expected, all the sequences cluster into subgroups according to their previously assigned classification (*Figure 1A*). Interestingly, mycobacterial Rv0435c, as well as its actinobacterial orthologs, form an independent cluster in the Cdc48-family tree (*Figure 1A*, blue branch). In contrast, sporadically

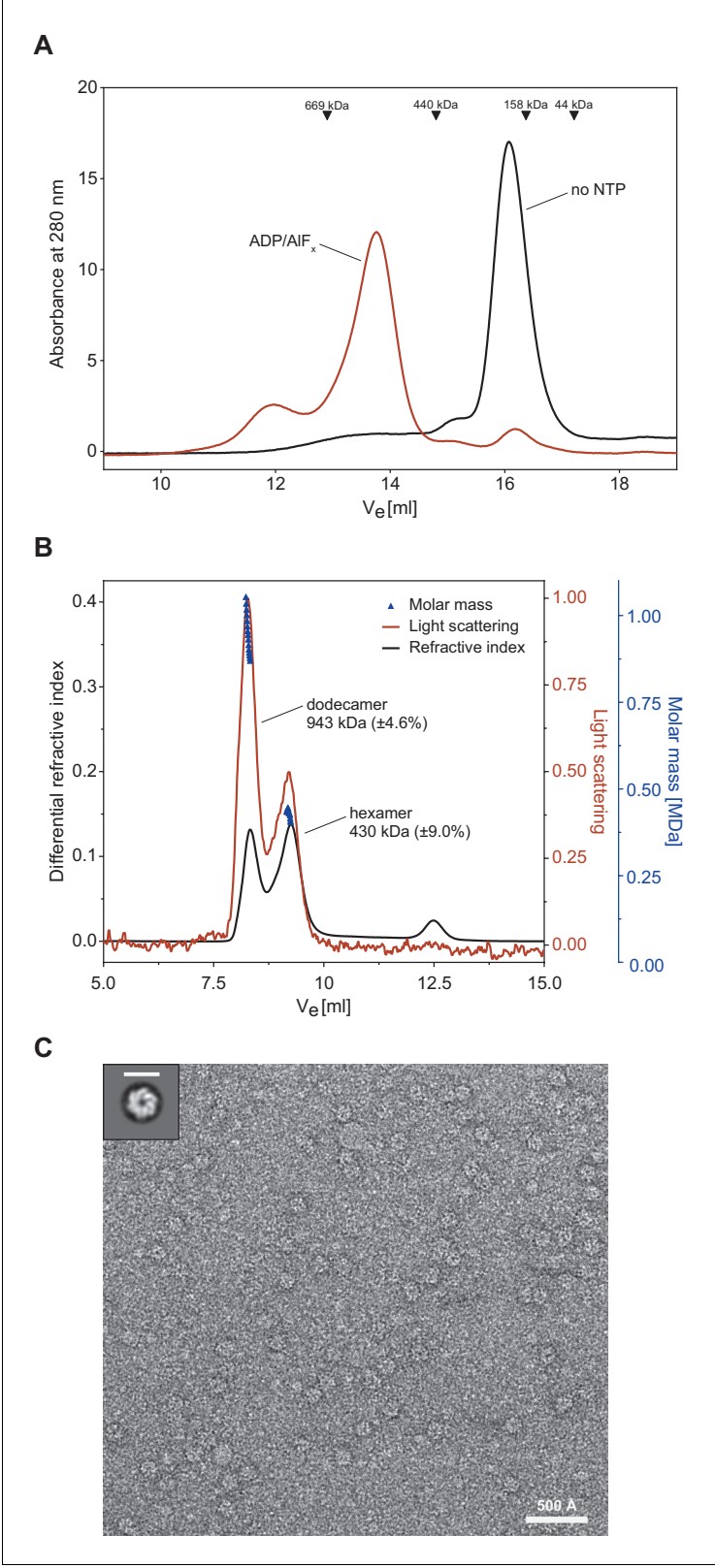

**Figure 2.** Cpa forms hexamers in presence of nucleotide. (**A**) Gel filtration profiles of 30 µM rhodococcal Cpa in absence of nucleotide (black line) and in presence of 2 mM ADP-AlF$_x$ (red line). The peak at ~16 ml corresponds to a monomer while the peak at ~13.5 ml corresponds to a hexameric assembly. (**B**) SEC-MALS profile of rhodococcal Cpa in presence of 2 mM ADP-AlF$_x$. The red line represents refractive index, the black line corresponds to the

*Figure 2 continued on next page*

*Figure 2 continued*

light scattering signal and dark-blue triangles represent the fitting of the molar mass. (**C**) A representative negative-staining electron micrograph of a double Walker B mutant of Cpa shows ring-shaped particles in the presence of ATP. Upper right panel shows a 2D class average of Cpa particles with a hexameric ring structure. Scale bar represents 150 Å.

DOI: https://doi.org/10.7554/eLife.34055.005

The following figure supplements are available for figure 2:

**Figure supplement 1.** Electron microscopy image averaging.

DOI: https://doi.org/10.7554/eLife.34055.006

**Figure supplement 2.** Mycobacterial Cpa forms hexamers in the presence of ADP-AlF$_x$ and readily hydrolyses ATP.

DOI: https://doi.org/10.7554/eLife.34055.007

occurring Cdc48 homologs of bacterial species outside of the phylum *Actinobacteria* do not cluster with their actinobacterial counterparts, but rather with archaeal Cdc48 family members (*Figure 1A*, grey branches within the green archaeal branch and lime-green group in *Figure 1—figure supplement 1B*). This, in combination with their infrequent occurrence in bacterial genomes would indicate that these bacteria acquired the *cdc48* gene by means of horizontal gene transfer from Archaea while the actinobacterial homolog Cpa is a descendant from a common Cdc48 ancestor.

To assess the possibility of Rv0435c acting in the context of the proteasome, we analyzed the occurrence of the *cpa* gene with respect to the proteasomal subunit genes *prcA* and *prcB* across the actinobacterial phylum (*Figure 1B*). Interestingly, none of the actinobacteria lacking proteasomal subunit genes encode a Cdc48 ortholog. In other words, where Cpa is present in actinobacteria, it co-occurs with the 20S proteasome. This observation supports the possibility that Cpa could be another proteasomal interactor, as it was shown for the eukaryotic and archaeal homologs (*Barthelme and Sauer, 2013*).

Analysis of the sequence alignment of 76 representative actinobacterial orthologs together with the recently solved crystal structure of the mycobacterial Cpa monomer show that Cpa features the canonical domain structure of tandem AAA-module proteins (*Unciuleac et al., 2016*). A small N-terminal domain (referred to as N-domain), usually responsible for binding adaptors and/or substrates, is followed by two consecutive canonical AAA modules (D1 and D2, respectively). To visualize the location of the domains and residues within a predicted hexameric ring, we built a structural homology model for the rhodococcal Cpa sequence using human p97 as a template, for which a structure of the assembled complex was available (*Hänzelmann and Schindelin, 2016*) (*Figure 1—figure supplement 1D*). The AAA modules each contain Walker A and B motifs required for ATP binding and hydrolysis; however, only the D2 module carries the so-called sensor asparagine (assisting the Walker B motif in coordinating water molecules) and arginine finger (stimulating the ATPase activity) (*Figure 1C* and *Figure 1—figure supplement 1D*) (*Unciuleac et al., 2016*). The alignment around the Walker A and B motifs shows strict conservation of the motifs in the D2 module, while the D1 module exhibits mild degeneration of the motifs (sequence logos in *Figure 1C*). To assess the sequence conservation more quantitatively and in context of the aforementioned Cdc48-like families, we employed the method described by *Wang and Kennedy (2014)* using principal component analysis to estimate the variance between the sequences being analyzed (*Figure 1—figure supplement 1A–C*). This analysis confirms that actinobacterial Cdc48-like proteins form a separate, tight cluster rather than clustering together with any specific other group (*Figure 1—figure supplement 1B*). Analysis of the second component loadings, which showcase the separation between the different families, revealed that the N-domains are the main distinguishing feature between the members of the different families, while the ATPase modules are highly conserved across the families (*Figure 1—figure supplement 1D*).

## Cpa forms hexameric rings upon nucleotide binding

Proteasomal activator complexes usually feature a ring-shaped architecture and associate with the 20S cylinder faces by a ring-stacking interaction aligning the activator ring pore with the 20S entry pore to form a conduit for substrate translocation. A crystal structure was recently determined for the *Mycobacterium smegmatis* (Msm) Cpa (Msm0858) in its monomeric form and the study

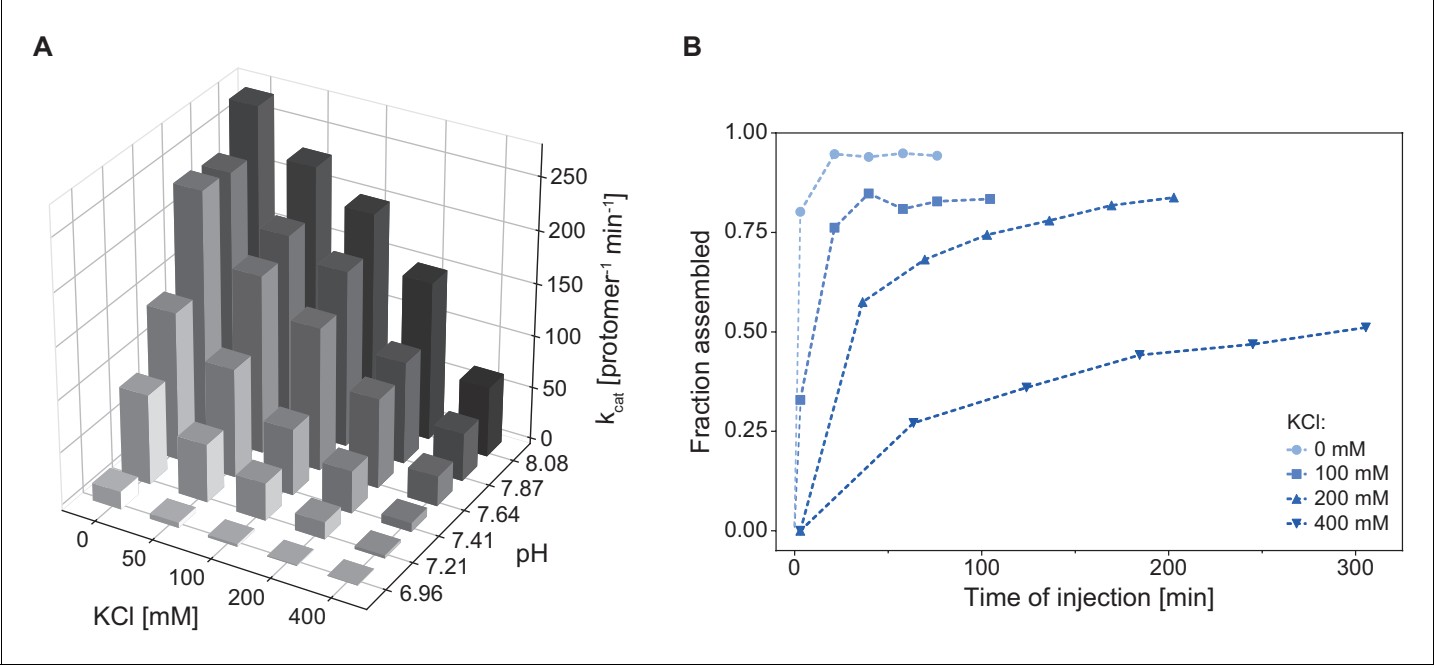

**Figure 3.** Cpa complex assembly and ATPase activity are strongly influenced by pH and ionic strength. (A) ATPase activity is enhanced with increasing pH and decreasing KCl concentration. The activity was measured at 28°C using 1.5 µM $^{Rer}$Cpa (protomer). For readability reasons, the errors of the measurements are not shown – in all cases the error did not exceed 5% of the measured value. (B) Increasing ionic strength slows down Cpa complex formation. Assembly reactions were prepared using 9 µM $^{Rer}$Cpa in a buffer containing increasing KCl concentrations (0–400 mM), incubated at 18°C and injected onto a SEC column at the indicated time points.

DOI: https://doi.org/10.7554/eLife.34055.008

concludes that Msm0858 does not form the canonical hexameric ring structures but rather acts as a monomer in solution (*Unciuleac et al., 2016*). As interaction with the 20S cylinder would require a ring-shaped complex, we hypothesized that actinobacterial Cdc48 homologs also assemble into hexamers like their eukaryotic and archaeal homologs. To perform size-exclusion experiments to test this hypothesis, we used recombinantly produced Cpa from *Rhodococcus erythropolis* due to its higher solubility compared to mycobacterial Cpa. In the absence of nucleotide, $^{Rer}$Cpa eluted in one main peak at an elution volume equivalent to a monomer/dimer assembly state (*Figure 2A*, black profile). Incubation of $^{Rer}$Cpa with the ATP-hydrolysis transition state analog ADP-AlF$_x$ prior to size exclusion resulted in a shift of the monomer/dimer elution peak to a position equivalent to just above 500 kDa (as indicated by the molecular weight standards), which is in good agreement with a hexameric assembly state (expected molecular weight: 450 kDa). An additional, smaller fraction elutes even earlier, at a position equivalent to larger than 700 kDa. In order to overcome the limitation of column calibration and obtain an accurate assessment of complex size, we also performed size exclusion chromatography with multi-angle light scattering detection (SEC-MALS). The light scattering signal revealed the mass of the assembled species in the main peak to correspond to 430 ± 39 kDa. This is in good agreement with the molecular weight of six monomers (75 kDa) in a hexameric assembly (*Figure 2B*). A fraction in *Figure 2A* eluted at an earlier position which corresponds to the molecular weight of a dodecameric assembly (943 kDa) and is likely due to the high protein concentration which was used for this experiment. To assess the shape of the assembled complex, we recorded electron micrographs of negatively stained Cpa particles. A double Walker B variant of the rhodococcal enzyme (D312N, E566Q) capable of nucleotide binding but not hydrolysis was used in this experiment to allow observation of the complex in the presence of ATP rather than ADP-AlF$_x$. The evenly distributed particles display as spherical shapes, most of which appear to be top views (*Figure 2C* and *Figure 2—figure supplement 1*). The 2D classification of the imaged particles revealed that most of the classes represent hexameric assemblies (*Figure 2—figure supplement 1*). Some of the particles, however, look slightly distorted, suggesting either high flexibility

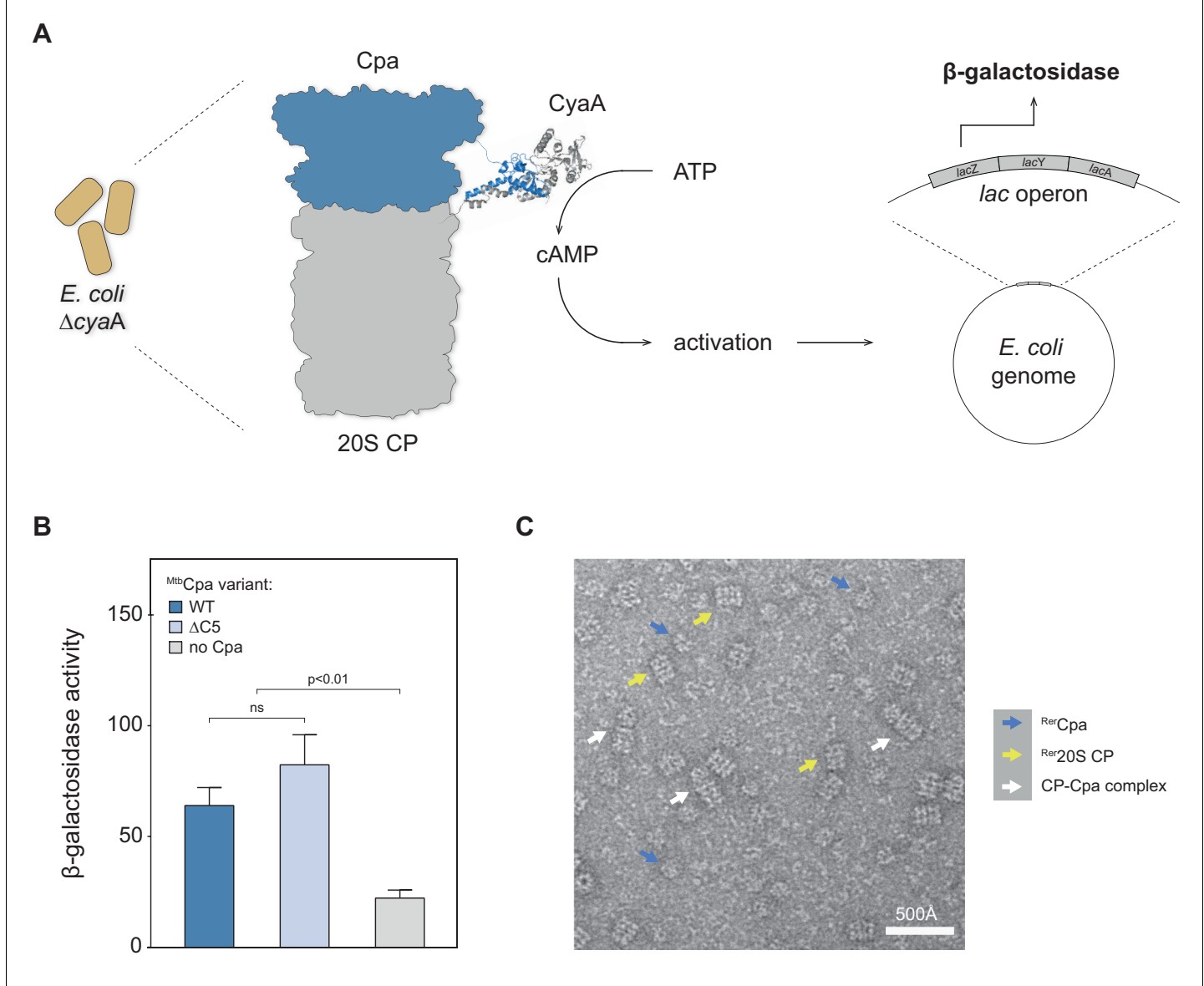

**Figure 4.** Cpa forms a complex with the 20S proteasomal core particle. (**A**) Principle of the bacterial adenylate cyclase two-hybrid system. The *cpa* gene was fused to the T25 subdomain of adenylate cyclase while the proteasome was fused to the adenylate cyclase T18 subdomain. Interaction of the two complexes in *E. coli* ΔcyaA gives rise to increased β-galactosidase activity. (**B**) Both MtbCpa wt as well as MtbCpa lacking the five C-terminal residues when coexpressed with CP produce an increase of β-galactosidase activity as compared to the negative control, suggesting Cpa-CP complex formation (statistical significance was tested using two-way ANOVA). (**C**) Electron micrograph of negatively stained Cpa particles and 20S proteasomes. White arrows indicate side views of stacked complexes between Cpa and the 20S proteasome, blue arrows point at free Cpa rings and yellow arrows indicate uncapped 20S CP side views.

DOI: https://doi.org/10.7554/eLife.34055.009

The following figure supplement is available for figure 4:

**Figure supplement 1.** Representative electron micrograph of negatively stained Cpa particles and 20S proteasomes.
DOI: https://doi.org/10.7554/eLife.34055.010

(presumably of the N-domains) or ring instability leading to ring opening in some of the 2D classes. Nevertheless, the diameter of the particles is rather uniform with roughly 150 Å and the appearance and dimensions of the particles agree well with hexameric ring assemblies that were observed previously for archaeal or eukaryotic Cdc48.

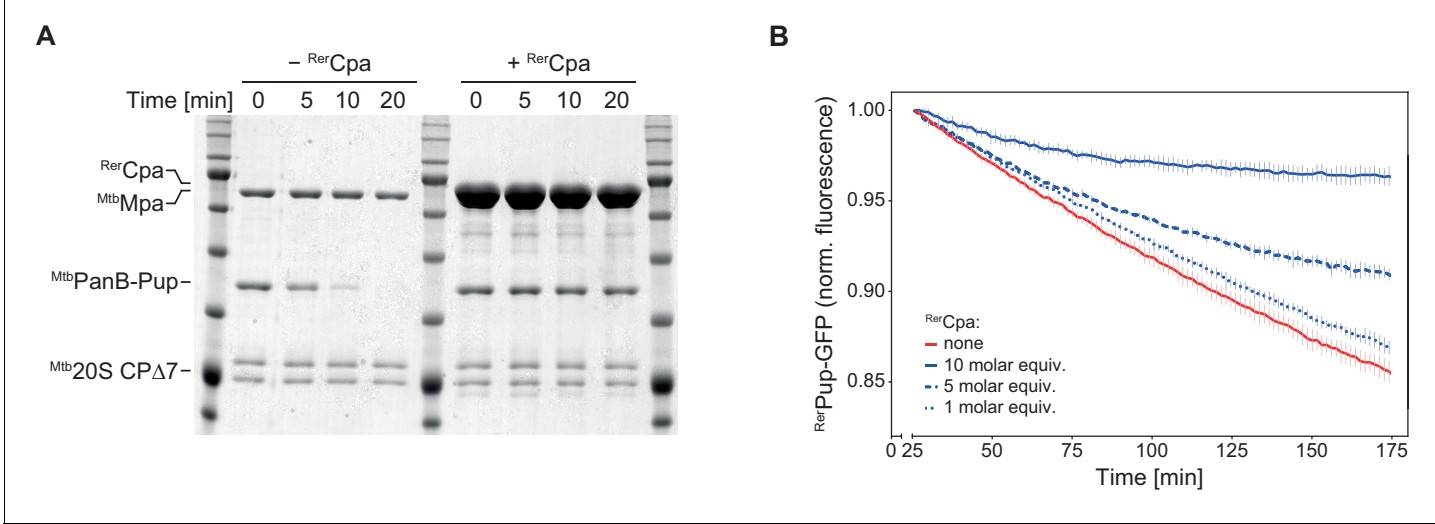

**Figure 5.** Cpa competes with Mpa for the binding site on the proteasomal core particle. (**A**) Rhodococcal Cpa slows down the degradation of PanB-Pup by mycobacterial Mpa-20S CP complex (concentrations: 4 μM [Mtb]PanB-Pup (protomer), 12 μM [Rer]Cpa (protomer), 0.2 μM Mpa (hexamer), 0.1 μM [Mtb]Δ7PrcAB complex). (**B**) Rhodococcal Cpa slows down the degradation of linear Pup-GFP fusion by rhodococcal ARC-20S CP complex in a concentration-dependent manner (concentrations: 0.5 μM [Rer]Pup-GFP, 0.6/1.5/3.0 μM [Rer]Cpa (protomer), 50 nM [Rer]Arc (hexamer), 25 nM [Rer]Δ7PrcAB complex).

DOI: https://doi.org/10.7554/eLife.34055.011

To test whether the mycobacterial Cpa can also form hexameric rings like the rhodococcal protein, we repeated the size-exclusion experiment described in the previous paragraph using [Msm]Cpa incubated in the presence of ADP-AlF$_x$ (*Figure 2—figure supplement 1*). The elution profile clearly shows that mycobacterial Cpa also forms hexamers in the presence of nucleotide. Additionally, we tested the ATPase activity of the mycobacterial enzyme at two different temperatures (28°C – temperature at which activity for the rhodococcal enzyme was determined, and 37°C – optimum growth temperature of Msm) (*Figure 2—figure supplement 2B*). Although mycobacterial Cpa exhibits low ATPase activity at 28°C, we observed a five-fold increase in activity when the temperature was increased to 37°C (change from 7.9 ± 2.9 to 42.5 ± 3.2 min$^{-1}$ hexamer$^{-1}$). Since the rhodococcal enzyme showed an overall better behaviour in vitro, subsequent in vitro experiments were performed using this homologue.

## ATPase activity of Cpa exhibits a strong dependence on pH and ionic strength

As demonstrated by the size-exclusion analysis, Cpa, like many of the AAA+ proteins that feature tandem AAA-modules, assembles into the hexamer only upon nucleotide binding. ATPase activity for these complexes is usually dependent on the formation of the assembled, hexameric ring (*Kress et al., 2007*). We therefore explored the dependence of ATPase activity on buffer conditions by varying pH and salt concentration, both factors known to affect assembly of AAA proteins. For example, ClpB/Hsp104 requires low-salt concentrations for assembly of the ring in vitro (*Wendler et al., 2012*). The rhodococcal Cpa exhibited a strong dependence of its ATPase activity on both, pH and KCl concentration, where increasing pH and decreasing salt concentration led to an increase in the measured enzyme activity by up to sevenfold (*Figure 3A*). We hypothesized that this strong dependence of the ATPase activity on salt was the result of enhanced ring formation under those conditions rather than a direct effect on the ATPase activity. To test this, we recorded assembly time courses at different salt concentrations (0–400 mM KCl) while keeping the pH constant at 7.8. Assembly was triggered by addition of Cpa into the assembly buffer at the given salt concentration and containing ADP-AlF$_x$ as nucleotide to avoid complications from ATP turnover. Aliquots were taken from the assembly reaction and subjected to size exclusion to determine the elution peak area of assembled Cpa. Interestingly, increasing the salt concentration from 0 to 400 mM led

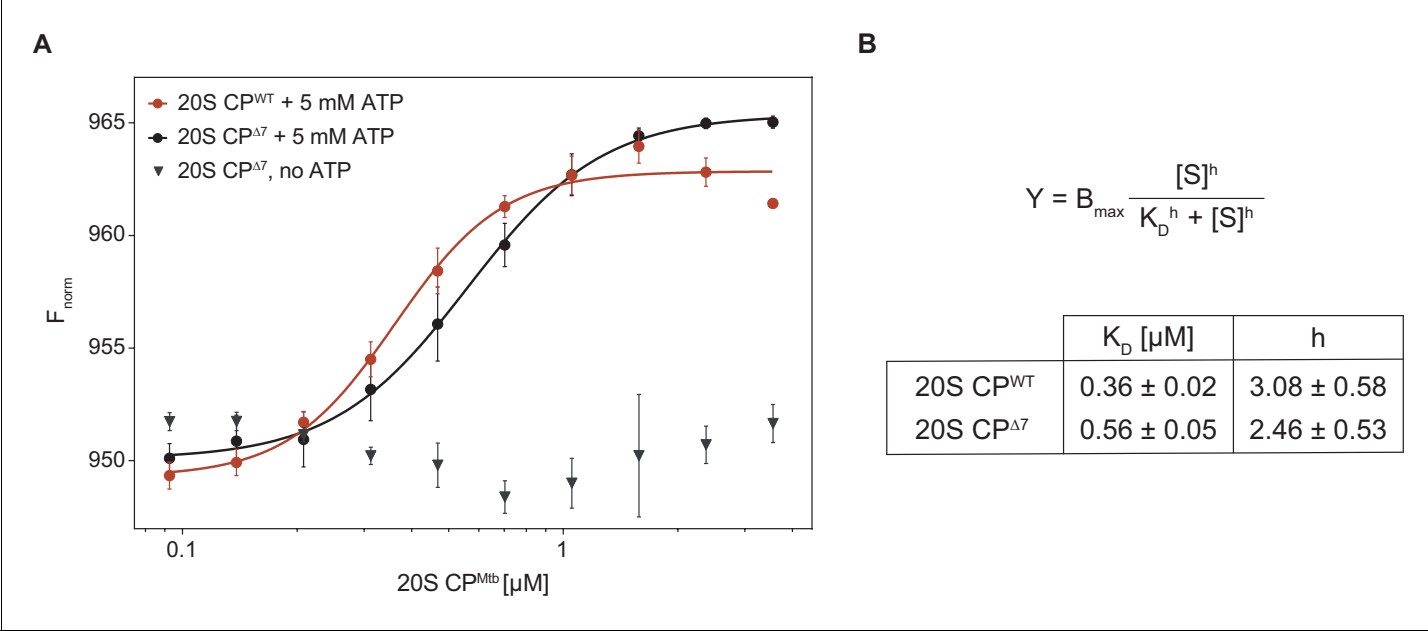

**Figure 6.** Microscale thermophoresis measurement of dissociation constant between [Rer]Cpa and [Mtb]20S CP. (**A**) Binding curves of [Rer]Cpa vs. closed-(red) and open-gate (black) [Mtb]CP in presence of ATP. The gray triangles represent a dataset recorded for Cpa and open-gate proteasome in absence of nucleotide. (**B**) Dissociation constants and Hill coefficients for the two measured datasets.

DOI: https://doi.org/10.7554/eLife.34055.012

to a strong deceleration of ring formation (*Figure 3B*). When no salt was present in the sample the oligomerization was complete within a few minutes, while at the highest KCl concentration only 40% of Cpa was assembled into rings after 5 hr of incubation. This result indicates that during the time frame in which the ATPase assay is performed the assembly is still slowly ongoing with its velocity dependent on the initial KCl concentration. Therefore, the measured activity reports mainly on the protein assembly state and indicates that ring formation is indeed strongly dependent on ionic strength.

## Cpa interacts with the 20S proteasome

Having established that Cpa is capable of hexameric ring formation, we were interested in testing whether the protein could interact with the 20S proteasome. To test for interaction between Cpa and the proteasome core particle, we employed the bacterial adenylate cyclase two-hybrid assay previously used for identifying interaction between the actinobacterial proteasome and another activator, Bpa (*Battesti and Bouveret, 2012*; *Delley et al., 2014*). The assay is based on the reconstitution of adenylate cyclase activity from the individually inactive T25 and T18 adenylate cyclase subdomains, which are tethered to the two potential interaction partners. In this case, Cpa from Mtb (Rv0435c) was fused to the T25 subdomain, while the T18 subdomain was fused to the α subunit of the 20S proteasome. Plasmids encoding the two respective fusion constructs were co-transformed into an *E. coli* strain deficient in adenylate cyclase activity. In case the two subdomain-carrying proteins, [Mtb]Cpa and [Mtb]20S proteasome, physically interact, the active adenylate cyclase complex can be reconstituted from the two subdomains, leading to production of cyclic AMP (cAMP) that in turn activates the *lac* operon (*Figure 4A*). To test for cAMP production, we therefore performed a β-galactosidase assay on cells permeabilized with SDS/chloroform. In addition to the wild type [Mtb]Cpa we also tested a variant where five C-terminal amino acids were removed. Although Cpa orthologs do not carry the classical HbYX proteasome interaction motif at their C-terminus, the C-terminus itself might nevertheless contribute to this interaction. *Figure 4B* shows the results of the β-galactosidase assay, including the negative (T25 against T18-CP) control. The activity of around 60 pmol 4-methylumbelliferone produced per min per OD unit indicates that the full-length mycobacterial Cpa is capable of directly interacting with the proteasomal core particle in the context of a bacterial cell.

Additionally, we observe that removal of the C-terminal amino acids does not change the interaction strength between Cpa and the 20S core particle, indicating that those residues do not significantly mediate this interaction. A milder contribution cannot be excluded on the basis of this assay due to the inherent positive feedback circuit that precludes precise quantification.

To probe for the interaction between Cpa and the proteasome in vitro, negatively stained EM micrographs of [Rer]Cpa rings pre-formed in the presence of ADP-AlF$_x$ and purified by size exclusion chromatography were recorded in presence of rhodococcal closed-gate proteasomes. We observed a small fraction of coaxial Cpa-proteasome complexes in side view (*Figure 4C* and *Figure 4—figure supplement 1*, white arrows) next to uncapped 20S cylinders (*Figure 4C*, yellow arrows) and free Cpa rings (*Figure 4C*, blue arrows).

Additionally, we tested the ability of [Rer]Cpa to compete with [Mtb]Mpa for complex formation with the proteasome. In this assay, degradation of a pupylated substrate by the [Mtb]Mpa-20S complex is observed in the absence or presence of an excess of [Rer]Cpa over [Mtb]Mpa. Competitive inhibition of the degradation reaction indicates complex formation between Cpa and the proteasome. In the absence of [Rer]Cpa, pupylated ketopanthoate hydroxymethyltransferase ([Mtb]PanB-Pup), a well-characterized degradation substrate of the [Mtb]Mpa-20S complex, is completely degraded within 20 min under the assay conditions, while in the presence of [Rer]Cpa no significant degradation takes place within the same time frame (*Figure 5A*). Using as a spectroscopically tracable model substrate a linear fusion of Pup to GFP (Pup-GFP), we also conducted the competition experiment with an all rhodococcal system (*Figure 5B*). Like in the heterologous setup, [Rer]Cpa could inhibit degradation of Pup-GFP by the [Rer]ARC-20S complex in a concentration-dependent manner, indicating interaction between Cpa and the 20S proteasome.

Finally, to quantify the interaction strength, we employed microscale thermophoresis (MST) to measure the dissociation constant between [Rer]Cpa and [Mtb]20S CP. For this purpose, we fused the small orange fluorescent protein mKO2 (monomeric Kusabira-Orange 2) to the N-terminus of [Rer]Cpa and performed titration experiments with closed- and open-gate mycobacterial proteasomes in presence and absence of ATP (*Figure 6*). In presence of ATP, we measured a dissociation constant of $0.36 \pm 0.02$ µM for the closed-gate proteasome and $0.56 \pm 0.05$ µM for the open-gate proteasome, indicating that the interaction between Cpa and closed-gate core particle is marginally stronger than with the open-gate variant. Importantly, we could not observe a signal change in absence of ATP, demonstrating that the measured interaction is specific to the Cpa hexamer that is formed only when nucleotide is present.

## Disruption of *cpa* in *Mycobacterium smegmatis* causes a growth defect under carbon starvation

In order to gain insight into the potential cellular function of mycobacterial Cpa, we generated an Msm *cpa* knockout strain by means of homologous recombination (Msm Δ*cpa*) and confirmed the absence of the protein using antibodies raised against [Msm]Cpa produced recombinantly in *E. coli*. (*Figure 7—figure supplement 1*). The knockout strain exhibited no growth defect compared to the parent strain under standard culture conditions (Middlebrook 7H9 medium with glycerol and Tween 80 at 37°C) (*Figure 7A*). However, transfer of the parent and knockout strain cultures grown in minimal medium supplemented with glycerol (adapted from (*Elharar et al., 2014*)) to a medium devoid of glycerol as the main carbon source reveals a growth defect in the knockout strain (*Figure 7B*). The wild-type Msm cells undergo two to three divisions during the first 72 hr after transfer, while the Msm Δ*cpa* cells stop dividing already after 24 hr. Once cell division has ceased, both strains can persist for at least 2 weeks and can be recovered when plated on 7H10 solid medium (data not shown). This result might point to a role of Cpa in adaptation of Msm to nutrient starvation stress.

## Cpa is encoded in an operon together with two other genes

Since bacterial operonic arrangement can give indications about the functional context in which a gene is required, we analyzed the genomic *cpa* locus across actinobacteria. As is evident from the phylogenetic tree shown in *Figure 1A*, the actinobacterial *cdc48* homolog cluster consists of three distinct branches based on their sequence alignment. Careful investigation of the genomic context in the three branches shows that in branches 1 and 2 (*Figure 1A and B*) *cpa* is encoded in a putative operon together with two other genes: *psd* (phosphatidylserine decarboxylase proenzyme) and

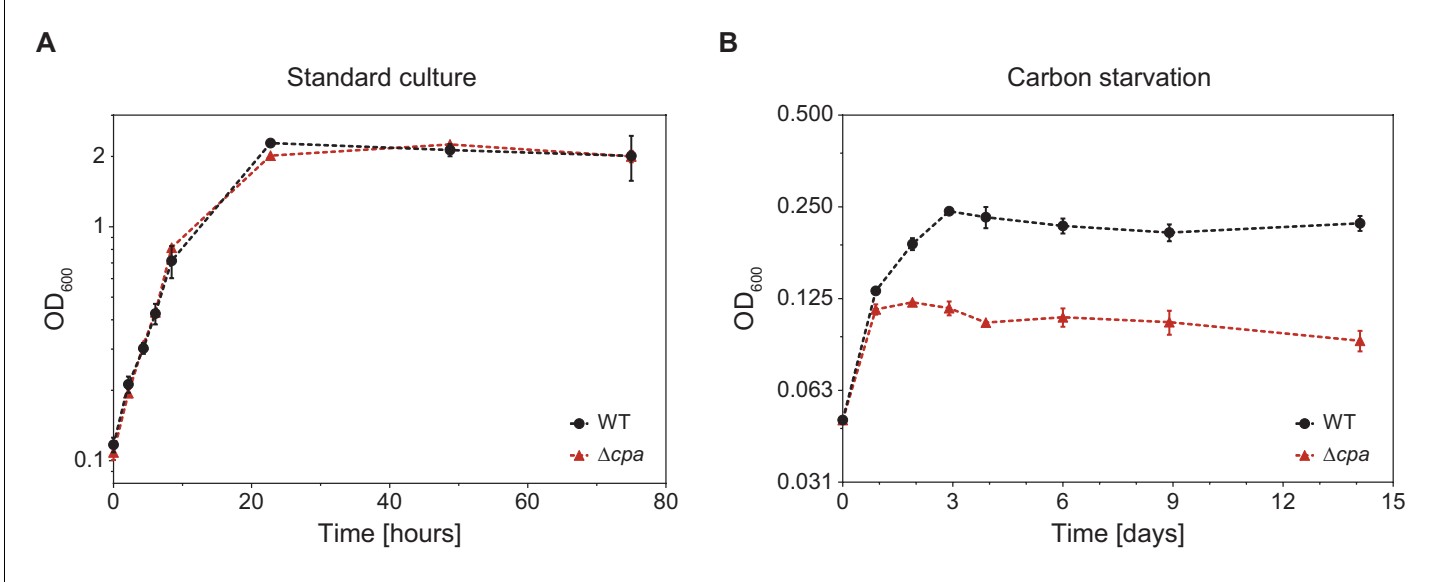

**Figure 7.** *M. smegmatis* Δ*cpa* shows normal growth behavior under standard culture conditions but displays a slight growth defect during carbon starvation. (**A**) Msm parent and knockout cells were cultured in Middlebrook 7H9 medium at 37°C and cell density was measured at 600 nm at the indicated time points. Both strains showed identical growth behavior indicating that Cpa is dispensable during standard cell culture conditions. (**B**) *cpa*-knockout cells show impaired growth in the same medium devoid of glycerol as a main carbon source. Both strains were cultured in a minimal medium in absence of glycerol at 37°C. Both growth curves are representative of three or more independent experiments with the mean values ± SD plotted.

DOI: https://doi.org/10.7554/eLife.34055.013

The following figure supplement is available for figure 7:

**Figure supplement 1.** Western blot analysis of the *Msm* Δ*cpa* strain.
DOI: https://doi.org/10.7554/eLife.34055.014

*pssA* (phosphatidylserine synthase), two consecutively acting enzymes responsible for the synthesis of phosphatidylethanolamine from CDP-diacylglycerol and serine. In the genomes of those two branches, the proteasomal genes are usually located within a relatively fixed distance from the *cpa* locus. In the actinobacteria belonging to the third branch, the *psd/pssA* genes exist, but are not located in the same operon as or in close proximity of the *cpa* gene. To probe if *psd*, *pssA* and *cpa* are co-transcribed, we designed six pairs of primers spanning different regions of the polycistronic RNA expected in case of co-transcription (*Figure 8A*). We used total RNA extracted from wild-type Msm to generate a single cDNA using a primer specific to the end of the *cpa* gene. In case the genes are transcribed from a single promoter placed upstream of the *psd* gene, we should generate all amplification products from the six primer pairs (labeled A-F in *Figure 8A*). Consistent with a single mRNA generated from the three genes, all six amplification products were present: two probes spanning joints between *psd/pssA* and *pssA/cpa* (probes A and B), a *cpa*-specific probe (probe C) as well as three longer probes, spanning two or three genes at the same time (probes D, E and F). Evidently, the *psd*, *pssA* and *cpa* genes are expressed together from a polycistronic operon.

Considering that phosphatidylethanolamine has been implicated in membrane curvature determination as well as cell division in some organisms (*Mileykovskaya et al., 1998*) and that the genes responsible for its synthesis are co-expressed with the *cpa* gene, we were interested to see whether removal of Cpa from the cell would influence the shape of starved cells. To test this, we grew Msm cultures of wt and *cpa*-knockout strains to the end of exponential phase (using medium with and without glycerol, as described above) and subsequently analyzed cell morphology using light microscopy (*Figure 8—figure supplement 1*). Since mycobacterial cells typically tend to form bigger cell aggregates in standard culture and individual cells are difficult to discern, we only inspected the cells visually without measuring individual cell lengths – no differences were apparent. Under carbon starvation, the cells did not form as many clumps and they were generally shorter compared to those grown in the presence of glycerol. To test whether the cells differ in length between the two strains we estimated the average cell length using light microscopy images. The knockout cells were longer

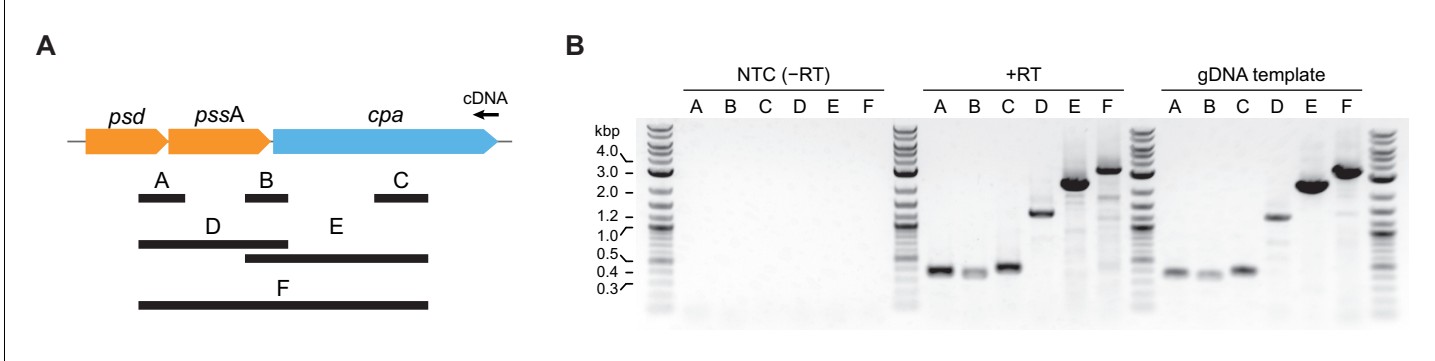

**Figure 8.** The *cpa* gene is co-transcribed together with neighboring *psd* and *pssA* genes. (**A**) Organization of the *cpa* gene locus in Msm and design of probes used to test for gene co-transcription. (**B**) Visualization of all six probes amplified from cDNA produced from total RNA using a *cpa*-specific primer (middle of the gel). A set of reactions without reverse transcriptase was included to test for presence of contaminating genomic DNA (NTC – no template control; left part of the gel) as well as standard PCR with genomic DNA as a template to visualize the expected length of all probes (right part of the gel).

DOI: https://doi.org/10.7554/eLife.34055.015

The following figure supplement is available for figure 8:

**Figure supplement 1.** *M. smegmatis* cells lacking *cpa* do not exhibit significant differences in size or shape to wild-type cells either under standard conditions or when grown in absence of glycerol.

DOI: https://doi.org/10.7554/eLife.34055.016

by approximately 13% compared to the wild-type cells. The difference, albeit statistically significant, seems too small to conclude that Cpa is involved in the regulation of cell division, although it cannot be excluded that Cpa could influence cell shape in a different way.

## Quantitative comparison of proteomic profiles of *M. smegmatis* wild-type with *cpa*-disrupted strains

In order to assess the effects of Cpa activity on the Msm proteome, we carried out differential proteomic analysis of the wild-type and Δ*cpa* Msm strains. To this end both strains were grown under standard culture conditions as well as under carbon starvation to the early stationary phase, when the cells were harvested and the soluble proteomes (enriched with membrane proteins; see Materials and methods section) of both strains determined using label-free quantification mass spectrometry (LFQ-MS). The relative protein changes together with their statistical significance are depicted in 'volcano' plots in *Figure 7A*. We observed 45 proteins during normal growth and 251 proteins during carbon starvation that were at least 1.5 times more abundant in the wild type strain compared to Msm Δ*cpa*. Likewise, 49 proteins were found to accumulate in the knockout cells during normal growth and 254 proteins during carbon starvation compared to Msm wt. Consequently, while during standard growth the differences are rather mild, exposing the cells to starvation stress led to significantly higher accumulation or depletion of a larger number of proteins in the *cpa* knockout strain. This observation strengthens the notion of Cpa playing a role during adaptation to nutrient-limited conditions. In order to gain a general understanding of the process in which Cpa may be involved directly, we classified the significantly changed proteins into functional classes using the Clusters of Orthologous Groups (COG) classification system (*Figure 7B*). We observed that in both cases (cells starved and non-starved), the class with most of the changed proteins was 'transcription'. Due to the low number of changes during standard growth, many COG classes under this condition were represented by only one or two proteins. This changed, however, when the cells were starved for carbon. We found that many changes occurred in classes like 'transcription' (most represented, similarly to standard growth), 'energy production and conversion', 'amino acid transport and metabolism' and 'lipid transport and metabolism'. From the perspective of Cpa acting in a protein degradation context, those proteins accumulating in the knockout strain could under the most simplistic interpretation be degradation substrate candidates. However, the fact that equal number of proteins accumulated as were depleted in the knockout strain suggests that this view is too simplistic. For example, degradation of a transcription factor that is present in the cell only with low abundance,

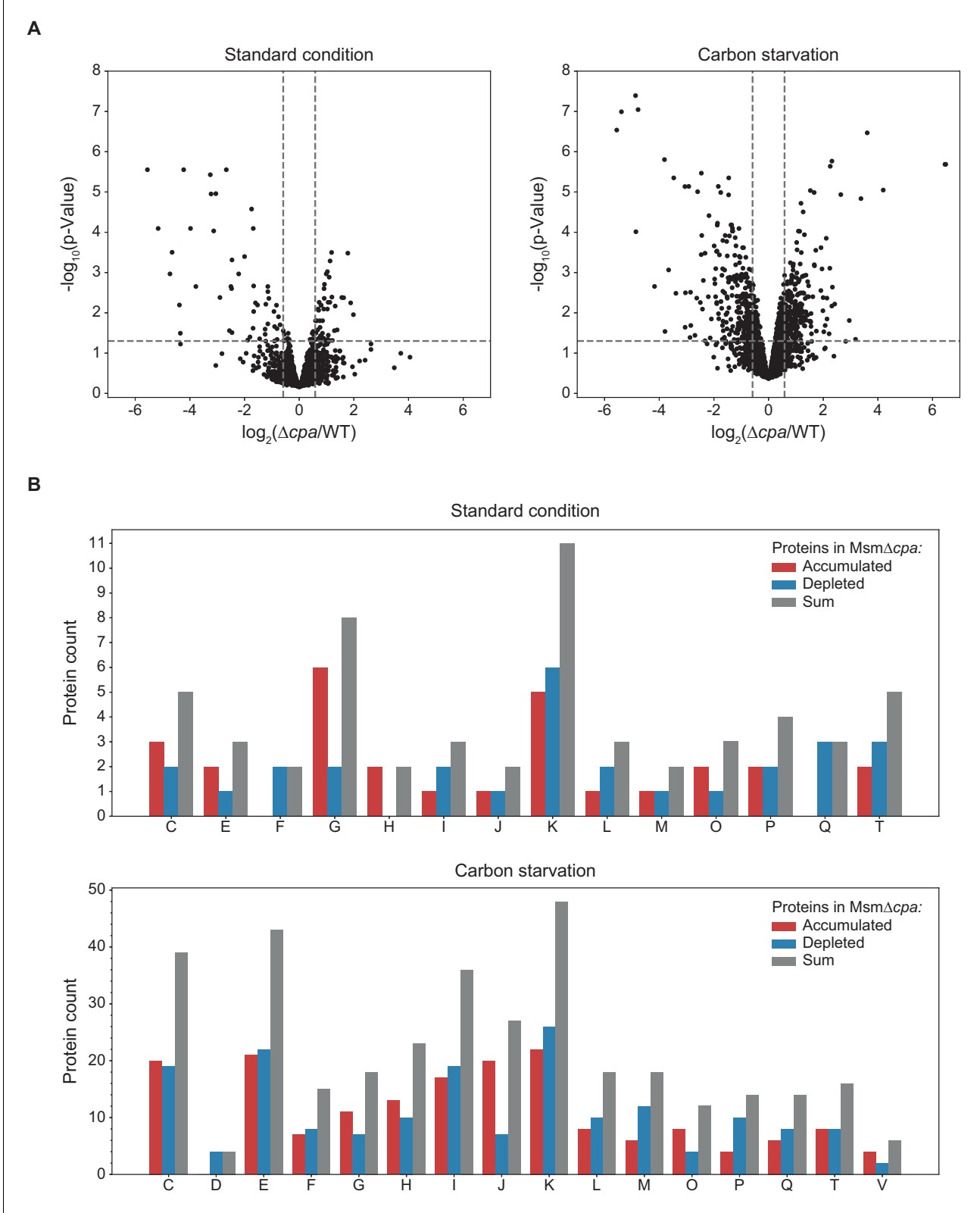

**Figure 9.** Label-free quantification mass spectrometry comparison of *M. smegmatis* WT with Δ*cpa* proteome. (**A**) Results of protein abundance comparison between the wt and the *cpa*-knockout strain using LFQ-MS (see *Figure 9—source data 1*). To filter the statistically-significant proteins, the p-value threshold was set to 0.05 (horizontal dashed line) and the fold-change threshold to 1.5 (vertical dashed lines). Under standard growth conditions (left plot), 49 proteins were found to accumulate (right side of the plot) and 45 proteins were found to be decreased (left side of the plot) in the

*Figure 9 continued on next page*

*Figure 9 continued*

knockout cells. Under carbon starvation (right plot), 254 proteins accumulated and 251 proteins were depleted in the knockout cells. (**B**) Those identified proteins for which a functional association was known or predicted, classification into functional classes using the COG classification system was carried out. Class abbreviations are as follows: C – energy production and conversion, D – cell cycle control and cell division, E – amino acid metabolism and transport, F – nucleotide metabolism and transport, G – carbohydrate metabolism and transport, H – coenzyme metabolism, I – lipid metabolism, J – translation/ribosomal structure and biogenesis, K – transcription, L – replication/recombination/repair, M – cell wall/membrane/envelope biogenesis, O – post-translational modification/protein turnover/chaperone functions, P – inorganic ion transport and metabolism, Q – secondary metabolites biosynthesis and catabolism, T – signal transduction, V – defense mechanisms. Proteins without an assigned class or assigned to class S (function unknown) were not included in the plot.

DOI: https://doi.org/10.7554/eLife.34055.017

The following source data and figure supplements are available for figure 9:

**Source data 1.** Source data file for *Figure 9*.
DOI: https://doi.org/10.7554/eLife.34055.020
**Source data 2.** Source data file for *Figure 9*.
DOI: https://doi.org/10.7554/eLife.34055.021
**Figure supplement 1.** STRING interaction network of the proteins that accumulated in the Cpa-knockout strain during carbon starvation.
DOI: https://doi.org/10.7554/eLife.34055.018
**Figure supplement 2.** Ribosomal proteins accumulating in the *cpa*-knockout cells during starvation mapped onto the structure of the 70S mycobacterial ribosome.
DOI: https://doi.org/10.7554/eLife.34055.019

could lead to significant proteomic changes. Indeed, as mentioned above, the most represented class in both our datasets is 'transcription', which could explain the observed changes in both directions.

To identify functional connections within the accumulating proteome, we performed a pathway enrichment analysis using STRING and DAVID tools (*Huang et al., 2009*; *Szklarczyk et al., 2017*). The result of this analysis showed an unusually high number of structural components of the ribosome (*Figure 9—figure supplements 1* and *2*) accumulating in the *cpa*-knockout cells. In fact, COG class J, encompassing proteins connected to translation and ribosome, was one of the very few classes that contained mostly proteins accumulating in the knockout cells and few with decreased levels (*Figure 7B*).

Finally, we were interested in testing whether some of the identified proteins could be Cpa binders in vivo under carbon starvation stress. For this purpose, we performed a co-immunoprecipitation (co-IP) experiment using an anti-<sup>Msm</sup>Cpa antibody and wild-type *M. smegmatis* cells (with Msm Δ*cpa* strain as a control for nonspecific binding to the beads/antibody). Using shotgun LC-MS/MS, we could identify a few dozen proteins in at least two out of three co-IP replicates that were specific to the presence of Cpa (i.e. did not bind to the antibody in the *cpa*-knockout lysate). We also submitted the results of the pull-down experiment to pathway enrichment analysis by STRING and DAVID tools and observed that again ribosomal proteins (indicated by bold rectangles in *Figure 9—figure supplement 2*) comprise a significantly enriched fraction of the immunoprecipitated protein mixture (see *Figure 9—source data 2* for the full list of identified binders).

## Discussion

Despite the vast amount of structural and functional information available for eukaryotic members of the Cdc48 family (p97 and NSF in particular), comparatively little is known about their prokaryotic homologs (*Bodnar and Rapoport, 2017*; *Kienle et al., 2016*). Only recently has it been established that archaeal equivalents of eukaryotic p97 are capable of directly interacting with the 20S proteasomal core and that they can support protein degradation in vitro (*Barthelme and Sauer, 2012*). The only previous study of an actinobacterial Cdc48-like ATPase is for the ortholog from Msm, and it reports that the protein exists as a monomer in solution as well as in the crystal structure (*Unciuleac et al., 2016*). Even though the authors suggest that the protein would need to undergo an oligomerization step in order to gain ATPase activity (due to ATP hydrolysis usually occuring at the interface between adjacent protomers), no evidence for assembly was presented.

In our study, we investigate different structural and functional aspects of Cdc48-like protein from actinobacteria (Cpa) using both rhodococcal and mycobacterial Cpa. In particular, the solubility and stability of the rhodococcal ATPase across a range of buffer conditions allowed us to study its oligomerization behavior by time-resolved size exclusion analysis and size exlusion chromatorgaphy coupled to a light scattering detection system. Our results clearly demonstrate that Cpa forms hexameric rings in the presence of nucleotide and that ATPase activity correlates with this assembly state. The most complete assembly into hexamers was observed in the presence of ADP-AlF$_x$, an analog of ATP that mimics the transition state of ATP hydrolysis, suggesting particular stabilization of the ring during ATP turnover. At least in vitro, the assembly into hexamers occurred most readily at low ionic strength, indicating that electrostatic interactions are crucial for rapid oligomerization. Another tandem AAA+ protein, the chaperone ClpB, can even assemble in absence of nucleotide in vitro if the ionic strength is kept low enough (*Schirmer et al., 2001*; *Schlee et al., 2001*). It is possible that like its eukaryotic counterparts, Cpa in vivo associates with a range of adaptor proteins that could further stabilize the ring. Such adaptor-dependent assembly behavior was for example shown for ClpC from *Bacillus subtilis* (*Kirstein et al., 2006*; *Schlothauer et al., 2003*).

The assembly of Cpa into a hexameric, ATPase-active ring complex, which is clearly demonstrated by our in vitro analysis, opens up the possibility that the Cpa ring can stack to the 20S proteasome α-rings, thereby forming a complex with centrally aligned ring pores. Importantly, the occurrence pattern of the *cpa* gene in actinobacterial genomes lends support to this argument, as *cpa* is found only in those members of actinobacteria carrying the two proteasomal subunit genes (*Figure 1B*). Genomic co-occurrence can indicate a functional connection between the co-occurring gene products, as is for example the case for components involved in the two known proteasomal degradation pathways in mycobacteria (*Barandun et al., 2012*; *Delley et al., 2014*). Furthermore, using a bacterial two-hybrid assay and electron microscopy we could demonstrate physical association between [Mtb]Cpa and the Mtb proteasome. This interaction occurs at the α-ring face of the proteasome, as can be seen in the negatively stained electron micrographs, and which is also supported by [Rer]Cpa competing with Mpa for binding to the proteasome in vitro, thereby inhibiting degradation of pupylated substrates. However, the molecular elements of this interaction differ from those employed by the other two actinobacterial proteasomal interactors, Mpa/Arc and Bpa, that both carry the canonical penultimate tyrosine motif, while Cpa does not. Interestingly, it has been shown for the archaeal and eukaryotic Cdc48 homologs, that an additional proteasome-interaction element exists that considerably contributes to the binding affinity (*Barthelme and Sauer, 2013*). Our attempt to test whether the removal of this molecular feature, the so-called D2 pore-2 loop, from Cpa would abolish its interaction with the 20S CP was unsuccessful due to the fact that removal or replacement of this element resulted in assembly defects of the hexameric ring. Nevertheless, it remains unclear, why the actinobacterial Cdc48 homolog does not carry both interaction motifs like its archaeal/eukaryotic cousins. One possible reason might be that formation of the Cpa-proteasome should be transient in nature. Alternatively, the different molecular arrangement around the actinobacterial α-ring of the proteasome might not be optimal for interaction with the Cpa C-terminus and therefore its contribution to the interaction minor or non-existent, so that the penultimate tyrosine was eventually lost. In fact, it would not be the only known case in the family of proteasomal regulators lacking the penultimate aromatic residue: the eukaryotic 11S (PA28) activator also does not carry the HbYX motif: instead, the C-terminal residue of each subunit provides proteasome binding energy and the so-called 'activation loops' trigger gate opening allowing for substrate entry (*Stadtmueller and Hill, 2011*; *Whitby et al., 2000*). However, our two-hybrid analysis showed that the last five C-terminal residues do not play a major role in proteasome binding, suggesting that instead other elements of Cpa must mediate the interaction.

Our in vivo analysis of a *cpa*-disrupted Msm strain suggests that Cpa plays a role during the adaptation to conditions where nutrients (carbon in particular) become limiting. In an attempt to better understand which aspects of cellular function Cpa might influence, we compared the proteomes of the parent Msm strain with the *cpa*-knockout strain grown in presence and absence of glycerol. Under both conditions, we identified proteins accumulating in the knockout strain as well as proteins with decreasing levels, indicating that the observed changes include not only potential degradation substrates but also proteins affected either through secondary effects like for example degradation of a regulatory protein or through Cpa functions independent of the proteasome. The proteins affected by the *cpa*-gene disruption fall into multiple functional categories, those involved in

transcription, energy production/conversion, amino acid metabolism/transport and lipid metabolism represented the most. Given the hampered growth of the knockout and the increased protein level changes under starvation conditions, these results suggest that Cpa could be involved in adaptation of the cell to a lack of nutrients. The proteins accumulating in the *cpa*-knockout strain potentially could comprise substrates that are usually removed with Cpa-involvement, either by direct degradation or by secondary effects. Our functional analysis of the proteome changes upon starvation (*Figure 9* and *Figure 9—figure supplement 1*) combined with identification of Cpa binders in the wild-type cells under the same conditions (*Figure 9—figure supplement 2*) could indicate that removal of Cpa from the cell more directly influences the level of ribosomal proteins compared to the other detected changes. In other bacteria it has been observed that nutrient stress (carbon starvation among others) results in ribosome disassembly and rRNA degradation (*Zundel et al., 2009*). Additionally, another recent study reported that an excess of ribosomal proteins is removed from yeast cells by the ubiquitin-proteasome system (*Sung et al., 2016*). It is tempting to hypothesise that actinobacterial Cpa is involved in a similar process; however, more effort will be required to investigate whether the role played by Cpa in disassembly and/or removal of ribosomal proteins under nutrient limitation requires cooperation with the proteasome core or is mainly mediated by its AAA pore-threading activity.

At the same time, it is difficult to speculate about the substrate clientele based on the homology of Cpa to eukaryotic Cdc48. For example, eukaryotic Cdc48 is implicated in processing of ubiquitinated proteins (*Richly et al., 2005*; *Rumpf and Jentsch, 2006*), a post-translational modification that does not occur in actinobacteria. Actinobacteria possess, however, the functionally analogous covalent modifier Pup that renders proteins as substrates for degradation by the Mpa-proteasome complex (*Striebel et al., 2010*; *Striebel et al., 2014*). The well-characterized pupylation substrate PanB in fact accumulates in the *cpa*-knockout strain. However, in vitro degradation tests with recombinantly produced and in vitro pupylated PanB showed no degradation in presence of Cpa and the proteasome (data not shown). Similarly, although the archaeal Cdc48-CP assembly is capable of degrading an ssrA-tagged model substrate in vitro, archaea do not harbor the tmRNA, a crucial element of trans-translation encoding the ssrA-tag (*Hayes and Keiler, 2010*; *Karzai et al., 2000*). Actinobacteria feature a tmRNA and carry out trans-translation. However, no Cpa-mediated degradation of ssrA-tagged model substrates could be detected in vitro (data not shown). As the Clp protease system is responsible for ssrA-tagged substrate degradation in mycobacteria and related bacteria, this finding was not unexpected (*Laederach et al., 2014*; *Raju et al., 2012*). The search for substrates is likely further hampered by the fact that Cdc48 family members almost always act in the context of adaptor proteins. In the absence of these adaptors, the activity toward the adaptor-mediated set of substrates is simply not detectable. The co-expression of Cpa from a polycistronic mRNA together with Psd and PssA, two consecutively acting enzymes responsible for the synthesis of phosphatidylethanolamine, might give an indication about the functional context of Cpa activity. Several studies indicate a functional link between phosphatidylethanolamine and cytokinesis, both in eukaryotes and bacteria (*Luo et al., 2009*; *Mileykovskaya et al., 1998*). This would suggest that Cpa might play a role in context of cell division. Removal of Cpa from the cells did not, however, show a significant effect on cell division as gauged by their cell size and morphology. Further investigations are clearly needed in order to gain a deeper understanding of the physiological role of Cpa in the biology of actinobacteria, including the search for adaptor proteins and analysis of their substrate recruitment clientele.

## Materials and methods

**Key resources table**

| Reagent type (species) or resource | Designation | Source or reference | Identifiers | Additional information |
|---|---|---|---|---|
| Gene | mKO2 | Thermo Fisher Scientific | NCBI accession: AB359188 | GeneArt synthetic construct |
| Strain (*Escherichia coli*) | E. coli Tuner(DE3) | EMD-Millipore | | |

*Continued on next page*

*Continued*

| Reagent type (species) or resource | Designation | Source or reference | Identifiers | Additional information |
|---|---|---|---|---|
| Strain (*Mycobacterium smegmatis* mc(2) 155) | Wild-type strain (Msm WT) | ATCC | ATCC: 700084 | |
| Genetic reagent (*Mycobacterium smegmatis* mc(2) 155) | Msm Δcpa | This paper | | Unmarked *cpa* deletion strain |
| Antibody | anti-RpoB[Ec] | BioLegend | BioLegend clone: 8RB13 | Antibody against beta-subunit of RNA polymerase from *E. coli*; used at dilution 1:1000 |
| Antibody | anti-Cpa[Msm] | This paper | | Rabbit polyclonal antibody against full-length Cpa from *M. smegmatis* mc(2) 155; produced by BioGenes GmbH; used at dilution 1:40000 |
| Recombinant DNA reagent | pET28a-His6-TEV-[Rer]Cpa | This paper | | Plasmid for expression of rhodococcal Cpa |
| Recombinant DNA reagent | pET28a-His6-TEV-[Msm]Cpa | This paper | | Plasmid for expression of mycobacterial Cpa |
| Recombinant DNA reagent | pET28a-His6-TEV-[Rer]Cpa_D312N_E566Q | This paper | | Plasmid for expression of double Walker B mutant of rhodococcal Cpa |
| Recombinant DNA reagent | pET28a-His6-TEV-mKO2-[Rer]Cpa | This paper | | Plasmid for expression of mKO2-Cpa fusion protein used in MST experiments |
| Recombinant DNA reagent | pETDuet-[Rer]PrcA-PrcBΔpro | This paper | | Plasmid for expression of rhodococcal proteasome (closed-gate variant) |
| Recombinant DNA reagent | pETDuet-[Rer]PrcAΔ7-PrcBΔpro | This paper | | Plasmid for expression of rhodococcal proteasome (open-gate variant) |
| Recombinant DNA reagent | pET28a-His6-TEV-[Rer]Arc | This paper | | Plasmid for expression of rhodococcal Arc |
| Recombinant DNA reagent | pETDuet-[Mtb]PanB-Strep | PMID: 20203624 | | |
| Recombinant DNA reagent | pTrc99-[Mtb]Mpa | PMID: 20203624 | | |
| Recombinant DNA reagent | pET20-His6-Trx-TEV-[Mtb]Pup-GFPuv | PMID: 20203624 | | |
| Recombinant DNA reagent | pKT25 | Euromedex | | |
| Recombinant DNA reagent | pUT18 | Euromedex | | |
| Recombinant DNA reagent | pKT25-zip | Euromedex | | |
| Recombinant DNA reagent | pUT18C-zip | Euromedex | | |
| Recombinant DNA reagent | pKT25-[Mtb]Cpa | This paper | | Bacterial two-hybrid plasmid: T25-Cpa fusion |
| Recombinant DNA reagent | pKT25-[Mtb]CpaΔC5 | This paper | | Bacterial two-hybrid plasmid: T25-Cpa fusion where Cpa is C-terminally truncated by five residues |

*Continued on next page*

*Continued*

| Reagent type (species) or resource | Designation | Source or reference | Identifiers | Additional information |
|---|---|---|---|---|
| Recombinant DNA reagent | pUT18-$^{Mtb}$PrcA-PrcBΔpro | This paper | | Bacterial two-hybrid plasmid: mycobacterial proteasome with the T18 subdomain fused to the C-terminus of PrcA |
| Commercial assay or kit | Monolith NT.115 Premium Coated Capillaries | NanoTemper Technologies | MO-K005 | |
| Chemical compound, drug | 4-Methylumbelliferone | Sigma-Aldrich | M1381 | |
| Chemical compound, drug | 4-Methylumbelliferyl β-D-galactopyranoside | Sigma-Aldrich | M1633 | |
| Software, algorithm | STRING | PMID: 27924014 | | |
| Software, algorithm | DAVID | PMID: 22543366 | | |
| Software, algorithm | FastTree | PMID: 20224823 | | |
| Software, algorithm | SWISS-MODEL | PMID: 28874689 | | |
| Software, algorithm | PyMOL | Schrödinger, LLC | http://www.pymol.org | |

## Alignment, phylogenetic tree construction and analysis of gene co-occurrence

All sequences were taken from the NCBI database (see *Supplementary file 1* for a full list of accession numbers) and aligned using the ClustalO multiple sequence alignment algorithm (*Sievers et al., 2011*). The resulting alignment was used to construct a phylogenetic tree with FastTree 2.1.1 (*Price et al., 2010*) applying default parameters. The results were visualized with FigTree 1.4.3.

Gene co-occurrence was investigated using the STRING tool (*Szklarczyk et al., 2017*) and verified by searching selected actinobacterial genomes for the presence of *bpa, pup-prcAB, arc, psd* and *pssA* genes with the help of BLAST (*Altschul et al., 1990*). Walker A/B sequence logos were generated from 76 Cpa (Cdc48-like protein of actinobacteria) sequences using WebLogo 2.8.2 (*Crooks et al., 2004*).

## Principal component analysis (PCA) of protein sequences

The multiple sequence alignment (described above) was used as an input for the principal component analysis as described by (*Wang and Kennedy, 2014*). Briefly, all the amino acid positions were ranked according to their occurrence at a given position and amino acids with the same occurrence were ranked according to their alphabetical order. The original residues were replaced with calculated ranks and the resulting table was used to perform PCA. All the calculations were carried out using a Python script (available at https://github.com/misialq/pca-protein-analysis; copy archived at https://github.com/elifesciences-publications/pca-protein-analysis).

A structural model of the rhodococcal Cpa was obtained using the SWISS-MODEL homology-modeling server where structure of the human p97 (PDB-ID: 5C1A) served as a template (*Biasini et al., 2014*). Sequence conservation was plotted according to the PCA loadings of the second principal component: positions whose PCA loadings exceeded 30% of the maximal PC2 loading were colored in red (variable) and those below 10% were colored in blue (conserved). Structure drawing was done in PyMOL 1.5 (Schrödinger, LLC).

## Bacterial strains and growth conditions

*Mycobacterium smegmatis* mc(2) 155 (*Msm*) was grown in liquid Middlebrook 7H9 medium (Difco) supplemented with 0.025% (w/v) Tween 80 and 0.2% (w/v) glycerol, unless otherwise stated. The cells were grown at 37°C with shaking at 180 rpm. For carbon starvation, the cells were first cultured

small-scale in minimal medium (adapted from (*Elharar et al., 2014*) containing: 40 mM $K_2HPO_4$, 22 mM $KH_2PO_4$, 15 mM $(NH_4)_2SO_4$, 1.7 mM sodium citrate, 0.4 mM $MgSO_4$, 0.05% (w/v) Tween 80 and 0.4% (w/v) glycerol. Washed cells from the small-scale culture were transferred into a large-scale culture in the same medium but lacking glycerol.

## Cloning, expression and protein purification

The genes encoding Rv0435c ($^{Mtb}$Cpa), MSMEG_0858 ($^{Msm}$Cpa) and RER_15150 ($^{Rer}$Cpa) were amplified from genomic DNA (*M. tuberculosis* – Mtb, *M. smegmatis* – Msm and *R. erythropolis* – Rer, respectively) by PCR using Q5 DNA polymerase (New England Biolabs; NEB) and cloned into a pET28a expression vector using Gibson assembly (NEBuilder HiFi DNA Assembly Kit; NEB). The double Walker B (D312N, E566Q) mutant of $^{Rer}$Cpa was cloned into the same backbone by Gibson assembly using gene pieces containing mutations introduced into PCR primers and amplified from rhodococcal genomic DNA. A His$_6$-tag followed by a Tobacco Etch Virus (TEV) protease cleavage site preceded the N-terminus of either protein. Constructs were transformed into *E. coli* Tuner(DE3) cells for heterologous expression which was carried out in ZYP-5052 autoinduction medium as described by *Studier (2005)* with minor modifications: the expression culture was inoculated to an $OD_{600} = 0.025$ using an overnight preculture and incubated at 30°C/180 rpm for 6 hr. After that time, the temperature was reduced to 18°C and the cells were harvested after an additional 14 hr. Cells were lysed in buffer A (50 mM HEPES-NaOH pH 7.8 at 8°C, 300 mM NaCl, 40 mM imidazole) using a high-pressure homogeniser (Microfluidics) and the cleared lysate was loaded onto $Ni^{2+}$-IMAC Sepharose 6 FF resin. After washing the resin with 15 column volumes of buffer A, the protein was eluted using buffer B containing 250 mM imidazole. Protein-containing fractions were pooled and dialysed for 30 min at 4°C against buffer C (50 mM HEPES-NaOH pH 7.8 at 8°C, 150 mM NaCl, 5 mM β-mercaptoethanol). The N-terminal hexahistidine tag was cleaved by addition of TEV protease and further dialysis for 14 hr. TEV protease was removed via $Ni^{2+}$ affinity chromatography, and Cpa was further purified by size exclusion chromatography using a Superdex 200 column in buffer C without β-mercaptoethanol. PrcAB, Arc/Mpa, PanB-Pup and Pup-GFP were purified as described previously (*Striebel et al., 2010*; *Striebel et al., 2009*).

To test for interaction between Cpa and the 20S proteasome using a bacterial adenylate cyclase two-hybrid assay (BACTH), the Mtb *cpa* full-length gene, as well as its variant with the last five C-terminal residues removed, was fused to the C-terminus of the T25 adenylate cyclase subdomain. The *cpa* gene (Rv0435c) was amplified from Mtb genomic DNA using Q5 DNA polymerase (NEB) and cloned into the pKT25 vector using Gibson assembly (NEB). The *prcAB* genes were cloned into the pUT18 vector where *prcA* was fused to the N-terminus of the T18 subdomain of adenylate cyclase. The *lac* promoter region was duplicated in this vector downstream of the *prcA*-T18 fusion followed by the *prcBΔpro* gene (propeptide residues 1–57 were removed). Empty pKT25 and pUT18 backbones, as well as T18-*zip* and T25-*zip* leucine zipper gene fusions (used as positive control), were obtained from Euromedex.

## Analytical size exclusion chromatography (SEC) and multi-angle light scattering (MALS)

Analytical SEC was performed on a Superose 6 Increase 10/300 GL column (GE Healthcare) connected to an Agilent 1260 Infinity HPLC system. All runs were performed at 23°C at a flowrate of 1 ml min$^{-1}$ in 50 mM HEPES-KOH pH7.8, 25 mM KCl, 20 mM $MgCl_2$. $^{Rer}$Cpa was injected at a concentration of 30 μM (protomer) in a volume of 15 μl and was detected by following absorption at 280 nm. Molecular weight standards were run under the same conditions. The sample with ADP-AlF$_x$ contained additionally 2 mM ADP, 2 mM $Al(NO_3)_3$ and 12 mM NaF.

For the assembly time trace, each sample was placed in the HPLC autosampler set to 18°C immediately after mixing and the injections onto the column were carried out at the indicated time points. Fraction of assembled protein was calculated by integrating all peaks and dividing the sum of the peak areas of hexamer and dodecamer by the sum of all peaks (hexamer, dodecamer and monomer).

To determine the molecular weight of the rhodococcal Cpa, 10 μl of 130 μM (protomer) protein solution were loaded onto a Superdex 200 Increase 10/300 GL column (GE Healthcare). The sample

was analyzed by refractive index using an Optilab T-rEX differential refractometer (Wyatt Technology) and by MALS using a miniDAWN TREOS light scattering detector (Wyatt Technology).

## Electron microscopy and image analysis

[Rer]Cpa-D312N,E566Q sample (300 nM) was incubated for 10 min at room temperature in the presence of ATP (2 mM) in 50 mM HEPES-KOH pH7.8, 25 mM KCl, 20 mM MgCl$_2$. After incubation, sample was diluted to final concentration of 75 nM and 8 µL were applied to Quantifoil grids R2/2 freshly coated with a thin layer of carbon and incubated for 2 min. Excess liquid was blotted away and the grids were stained with 2% uranyl acetate for 2 min. Images were collected on FEI F20 electron microscope operated at 200 keV equipped with Faclon II direct electron detector (FEI Company) at x 82,350 magnification with a dose of 30 electron per Å$^2$, defocus value of $-2$ µm, and a pixel size of 1.7 Å per pixel. Particles were selected semi-automatically with BOXER implemented in the EMAN package (*Ludtke et al., 1999*). Particles (17,375) were extracted using RELION (*Scheres, 2012*) and binned three-fold resulting in a pixel size of 5.1 Å per pixel. Two-dimensional (2D) maximum-likelihood classification was performed with the RELION package (*Scheres, 2012*) using K = 30 and was run for 15 iterations.

To image Cpa-proteasome interaction, 75 nM [Rer]Cpa rings in the presence of ADP-AlF$_x$ and 30 nM 20S closed-gate rhodococcal proteasome were incubated at room temperature in 50 µl reactions and then applied to Quantifoil grids R2/2 freshly coated with a thin layer of carbon followed by staining with 2% uranyl acetate similarly as described above. The samples were then imaged using a FEI Morgagni 268 transmission electron microscope operating at 100 keV.

## ATPase activity assay

ATPase activity was determined using an ATPase assay coupled to pyruvate kinase and lactate dehydrogenase (*Nørby, 1988*). Reactions were performed in 96-well plates at 28°C (or 37°C for [Msm]Cpa), and absorption at 340 nm was monitored using a Synergy 2 plate reader (BioTek). The assay was performed in 50 mM HEPES-KOH, 2.5 mM phosphoenolpyruvate, 1 mM NADH, 40 U ml$^{-1}$ of both pyruvate kinase and lactate dehydrogenase, 20 mM MgCl$_2$ and 2 mM DTT, where pH was varied in the range of pH 7 to pH 8 and KCl was varied from 0 to 200 mM. The total reaction volume was 100 µl, and the assay was carried out with 1 mM ATP and 1.5 µM [Rer]Cpa (or 3 µM [Msm]Cpa) enzyme. The rate of ATP hydrolysis was calculated using an extinction coefficient of 6.22 mM$^{-1}$ cm$^{-1}$ at 340 nm for NADH.

## Protein interaction analysis using BACTH

To test the interaction between [Mtb]PrcAB and [Mtb]Cpa, a bacterial adenylate cyclase two-hybrid (BACTH) assay was performed as described by Battesti and Bouveret with some modifications (*Battesti and Bouveret, 2012*). Briefly, *E. coli* BTH101 Δ*cya* cells were co-transformed with the pKT25 and pUT18 fusion constructs described above. As a negative control, bacteria were co-transformed with empty pKT25 vector. Co-transformants were plated on LB agar plates supplemented with 100 µg ml$^{-1}$ ampicillin and 50 µg ml$^{-1}$ kanamycin and incubated for 72 hr at 30°C. Several clones were grown for 20 hr at 30°C in liquid LB medium containing both antibiotics and 0.5 mM IPTG. 100 µl of the overnight cultures were transferred into 400 µl of buffer Z (44.7 mM Na$_2$HPO$_4$, 45.3 NaH$_2$PO$_4$, 10 mM KCl, 1 mM MgSO$_4$, 38.5 mM β-mercaptoethanol; pH 7), followed by addition of 25 µl 0.01% SDS and 50 µl chloroform and vigorous mixing. 20 µl of each cell extract were transferred to a 96-well nonbinding plate (Corning), and the β-galactosidase reaction was started by addition of 30 µl 0.83 mM 4-methylumbelliferyl β-D-galactopyranoside (Sigma-Aldrich). After 20 min of incubation at room temperature, the reactions were stopped by addition of 50 µl 1 M Na$_2$CO$_3$ and the fluorescence was measured in a Synergy 2 plate reader (BioTek) using a 360/40 excitation filter and a 460/40 emission filter. The amount of released 4-methylumbelliferone was determined using a standard curve prepared the same day (Sigma-Aldrich). The β-galactosidase activity was normalized to OD$_{600}$ of the overnight cultures used for the assay. The final activity is expressed as pmol of 4-methylumbelliferone min$^{-1}$ OD unit$^{-1}$.

## Proteasome competition assay

The gel-based competition assay with Mpa was performed as described previously (*Delley et al., 2014*) with minor modifications. $^{Mtb}$PanB-Pup (4 μM protomer) was pre-incubated for 15 min at 30°C with $^{Mtb}$Δ7PrcAB (0.1 μM complex), 5 mM ATP, 40 mM phosphocreatine, 0.5 U ml$^{-1}$ creatine phosphokinase and 1 mM DTT in buffer M2 (50 mM HEPES-KOH pH 7.5 at 23°C, 100 mM KCl, 20 mM MgCl$_2$, 10% glycerol) in presence or absence of $^{Rer}$Cpa (12 μM protomer). The degradation reaction was started by addition of 0.2 μM Mpa (hexamer). Aliquots were withdrawn at the indicated time points, and the reactions were stopped by addition of Laemmli sample buffer. The aliquots were analyzed by SDS-PAGE followed by staining with Coomassie Brilliant Blue.

For the assay with a fluorescent GFP readout, $^{Rer}$Pup-GFP (0.5 μM) was pre-incubated for 15 min at 30°C with $^{Rer}$Δ7PrcAB (25 nM complex), 5 mM ATP, 40 mM phosphocreatine, 0.5 U ml$^{-1}$ creatine phosphokinase and 1 mM DTT in buffer M2 in presence or absence of $^{Rer}$Cpa (0.6, 1.5 or 3 μM protomer). The degradation reaction was started by addition of 50 nM $^{Rer}$Arc (hexamer). The decrease in fluorescence (Ex$_{485/20}$, Em$_{528/20}$) was monitored using a Synergy 2 plate reader (BioTek) at 28°C.

## Microscale thermophoresis (MST)

For MST experiments, rhodococcal Cpa was N-terminally fused to the orange fluorescent protein mKO2 (monomeric Kusabira-Orange 2), expressed and purified as described for the wild-type Cpa. 20S mycobacterial proteasomes (closed- and open-gate variant) were titrated to 50 nM mKO2-$^{Rer}$Cpa using a 1.5x dilution series between 0.09 and 3.5 μM in 50 mM HEPES-KOH, 50 mM NaCl, 0.005% Tween 20 (and 5 mM ATP, 20 mM MgCl$_2$ in the samples containing ATP) in a total of 16 μl. After 5 min of incubation at room temperature, the samples were transferred to Monolith NT.115 premium coated capillaries (NanoTemper Technologies) and MST traces were recorded at 25°C in a Monolith NT.115 instrument (NanoTemper Technologies; LED power 60%, MST power 20%). All measurements were performed in three independent replicates.

The MST data were fitted using the T-Jump section of the traces using the Hill equation: $y = U + (B - U)(S^h/(K_d^h + S^h))$, where U is the signal from the unbound state, B is the signal from the bound state, S is the concentration of the titrated species, $K_d$ is the dissociation constant and h is the Hill coefficient. Fitting was performed using GraphPad Prism 7.

## Disruption of *cpa* (MSMEG_0858) in *M. smegmatis* mc(2) 155

A targeted allelic exchange of the MSMEG_0858 gene was performed by application of a two-step selection with the p2NIL/pGOAL system (*Gopinath et al., 2015*). First, the competent *Msm* WT cells were transformed with the suicide knockout plasmid pΔ*cpa* and plated on 7H10 medium supplemented with 20 μg ml$^{-1}$ kanamycin and 50 μg ml$^{-1}$ hygromycin B to select for single-crossover mutants. An 'X-Gal underlay' was performed to visualize the clones that were successful recombinants: 200 μl of 0.4% X-Gal solution were pipetted under the agar, and the plate was incubated for an additional 1–2 days. Blue clones were used to inoculate fresh Middlebrook 7H9 medium containing kanamycin and hygromycin and incubated for several days at 37°C. To select for double-crossover mutants, the cells were plated onto Middlebrook 7H10 agar medium supplemented with 2% sucrose and control medium without sucrose and incubated at 37°C for several days. The 'X-gal underlay' was repeated and several sucrose-resistant white clones were picked and re-plated onto fresh Middlebrook 7H10 agar plates with and without kanamycin. Several kanamycin-sensitive clones were picked, and absence of the *cpa* gene was confirmed by colony PCR using OneTaq polymerase (NEB) and primers flanking the disruption site. The resulting mutant strain (referred to as Msm Δ*cpa*) was additionally confirmed by PCR amplification of the whole *psd-pssA*-Δ*cpa* locus using Q5 polymerase (NEB) followed by sequencing of the resulting fragment, as well as western blot detection of the protein product using a rabbit polyclonal antibody (Biogenes, Germany) raised against recombinant $^{Msm}$Cpa expressed and purified from *E. coli*.

## Test for co-transcription of *psd*, *pssA* and *cpa*

Total RNA isolated from Msm SMR5 was used as a template for synthesis of cDNA with a primer specific to the 3' end of the *cpa* mRNA. cDNA synthesis was performed with 200 U of Maxima Reverse Transcriptase (Thermo Fisher Scientific) in the presence of 20 U of the RiboLock RNase Inhibitor (Thermo Fisher Scientific) using a step-wise temperature gradient: 50°C for 30 min, 55°C for

15 min and 65°C for 30 min. The resulting cDNA was used directly in a PCR reaction with OneTaq polymerase (NEB) using six primer pairs for amplification of different regions of the transcript: two pairs for joints between *psd/pss*A and *pss*A/*cpa* genes, one *cpa*-specific pair, two pairs spanning more than one gene and one pair spanning all three genes of interest. To control for genomic DNA contamination, a set of samples was prepared where the reverse transcriptase was omitted. Additionally, to check the expected correct size of all the fragments, PCR reactions were also performed directly with genomic DNA where no RNA was present. PCR products were later analyzed using a 1% agarose gel and visualized in the presence of ethidium bromide.

## Label-free quantification mass spectrometry

To obtain information about the relative proteome changes in the wild-type compared to the Δ*cpa* Msm strain we used the label-free quantification mass spectrometry approach. For this purpose WT and knockout cells were grown in pentaplicates in minimal medium (described above) in a similar way as during growth curve determination, to an $OD_{600} = 2$ for cultures with glycerol and $OD_{600} = 0.12$–$0.25$ for cultures without glycerol. Cells were harvested by centrifugation and washed three times with 1x PBS. Cell pellets were resuspended in 1 ml of 1x PBS/2 mM EDTA (containing 1x Halt Protease Inhibitor Cocktail; Thermo Fisher Scientific) and transferred to 2 ml screw-cap tubes containing ~ 0.5 g of 0.15 mm zirconium oxide beads (Next Advance). The cells were lysed by bead-beating two times for 30 s at maximum speed with a 1 min cooling pause in a Minilys homogeniser (Bertin Instruments). Intact cells were removed by centrifuging twice at 3000 g at 4°C for 15 min. Resulting supernatants were transferred to fresh tubes and membrane fractions were recovered by centrifugation at 100000 g at 4°C for 4 hr. The supernatant (soluble fraction) was transferred to a new tube and the pellet was gently washed with 200 μl of 1x PBS/2 mM EDTA and spun down again at 100000 g at 4°C for 50 min. Subsequently, washed pellets were resuspended in 200 μl of 50 mM $NH_4HCO_3$ by incubation in an ultrasonic bath for 1 hr (SONOREX 10 P, 70% power, 4°C). Membrane proteins were solubilised by addition of 250 μl of 2% SDS (final concentration of 1.1%) and an overnight incubation at room temperature. Insoluble material was spun down at 21000 g for 30 min at room temperature and the pellet was resuspended in 2% SDS. After an additional 1 hr incubation, the sample was centrifuged again and the supernatant was combined with the supernatant from the previous step (solubilised membrane proteins). The proteins were precipitated by addition of 5 volumes of ice-cold acetone, followed by an overnight incubation at −20°C. Precipitated proteins were collected by centrifugation at 21000 g at 4°C for 30 min, the pellets were air-dried and resuspended in 100 μl of 3 M urea. Protein concentration was determined using the BCA method with bovine serum albumin (BSA) as a standard. The soluble fraction was mixed with the membrane fraction in a ratio 2:1 for a total concentration of 0.75 mg ml$^{-1}$. Equal amounts of protein were analysed by SDS-PAGE followed by immunoblot with an antibody against $^{Msm}$Cpa (see above) or $^{Ecoli}$RpoB as a loading control. Cell lysates were submitted for LFQ-MS analysis to the Functional Genomics Center Zurich (FGCZ).

## Co-immunoprecipitation (co-IP) using anti-Cpa antibody

For the α-Cpa co-IP experiment, 100 ml cultures of Msm wild-type and Δ*cpa* cells were grown in absence of glycerol as described in the previous section. Following harvesting, cell pellets were resuspended in 1 ml of 1x PBS/2 mM EDTA and lysed as described above. The lysates were then spun down at 20000 g at 4°C for 10 min and the supernatants were kept on ice. Anti-$^{Msm}$Cpa antibody was immobilized on 50 μl (per sample) Dynabeads$^{TM}$ Protein A (Thermo Fisher Scientific) and crosslinked using BS$^3$ crosslinker, following manufacturer instructions. The beads were then mixed with 600 μl of the lysate, supplied with 5 mM ATP and 15 mM $MgCl_2$ and incubated at 30°C for 15 min with gentle agitation. Next, the beads were washed three times with 200 μl of 1x PBS/5 mM ATP/15 mM $MgCl_2$ and the immobilized proteins were eluted with 50 μl of 0.2 M glycine pH 2.1 and immediately neutralized by addition of 10 μl of 1 M NaOH. 25 μl of each elution were analyzed using shotgun LC-MS/MS at the Functional Genomics Center Zurich (FGCZ). Co-immunoprecipitations were performed in triplicate.

## Acknowledgements

This work was supported by the Swiss National Science Foundation. We thank Paolo Nanni, Claudia Fortes and Laura Kunz from the FGCZ proteomics team for the LFQ-MS analysis, technical support and advice. We thank Andreas Müller for providing a sample of mycobacterial RNA for the co-transcription experiment and Kasia Radomska for help with light microscopy. We are grateful to Nenad Ban and Daniel Böhringer for helpful discussions concerning the EM analysis.

## Additional information

### Funding

| Funder | Grant reference number | Author |
| --- | --- | --- |
| Schweizerischer Nationalfonds zur Förderung der Wissenschaftlichen Forschung | 163314 | Eilika Weber-Ban |

The funders had no role in study design, data collection and interpretation, or the decision to submit the work for publication.

### Author contributions

Michal Ziemski, Conceptualization, Formal analysis, Investigation, Visualization, Writing—original draft, Writing—review and editing; Ahmad Jomaa, Formal analysis, Investigation, Prepared EM grids and performed negative-stain EM, Collected and analyzed the EM data; Daniel Mayer, Formal analysis, Investigation, Helped with the MALS measurements; Sonja Rutz, Investigation, Helped with initial growth experiments; Christoph Giese, Investigation, Prepared EM grids and performed negative-stain EM; Dmitry Veprintsev, Resources, Helped with the MALS measurements; Eilika Weber-Ban, Conceptualization, Resources, Supervision, Funding acquisition, Writing—review and editing

### Author ORCIDs

Michal Ziemski http://orcid.org/0000-0001-6285-8852
Eilika Weber-Ban http://orcid.org/0000-0002-5773-9274

### Decision letter and Author response

Decision letter https://doi.org/10.7554/eLife.34055.027
Author response https://doi.org/10.7554/eLife.34055.028

## Additional files

### Supplementary files

• Supplementary file 1. Supplementary tables. Supplementary Table 1. List of proteins used for ClustalO alignment of the Cdc48 family (NCBI accession numbers). Supplementary Table 2. List of proteins accumulating during growth in the presence of glycerol in *M. smegmatis* Δcpa as compared to its parent strain by label-free quantification mass spectrometry. Supplementary Table 3. List of proteins depleted during growth in the presence of glycerol in *M. smegmatis* Δcpa as compared to its parent strain by label-free quantification mass spectrometry. Supplementary Table 4. List of proteins accumulating during growth in the absence of glycerol in *M. smegmatis* Δcpa as compared to its parent strain by label-free quantification mass spectrometry. Supplementary Table 5. List of proteins depleted during growth in the absence of glycerol in *M. smegmatis* Δcpa as compared to its parent strain by label-free quantification mass spectrometry.
DOI: https://doi.org/10.7554/eLife.34055.022

• Transparent reporting form
DOI: https://doi.org/10.7554/eLife.34055.023

**Data availability**

All data generated or analysed during this study are included in the manuscript and supporting files. Source data files have been provided for Figure 9 and Supplementary Figures 7 and 8.

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
