## [Decision Letter]

Thank you for submitting your article "Cdc48-like protein of actinobacteria (Cpa) is a novel proteasome interactor in mycobacteria and related organisms" for consideration by *eLife*. Your article has been reviewed by 3 peer reviewers, one of whom was the Reviewing Editor, and the evaluation has been overseen by Andreas Martin as the Reviewing Editor and Ivan Dikic as the Senior Editor. The following individual involved in review of your submission has agreed to reveal their identity: Andreas Martin (Reviewer #1); Andreas Matouschek (Reviewer #2).

The reviewers have discussed the reviews with one another and the Reviewing Editor has drafted this decision to help you prepare a revised submission.

Summary:

This manuscript provides an important advance to our understanding of mycobacterial proteasome interactors by showing that the Cdc48-like protein of actinobacteria (Cpa) forms hexameric rings, hydrolyzes ATP, and may functionally bind to the 20S core peptidase. Furthermore, changes in the levels of ~ 500 proteins upon Cpa knockout in *M. smegmatis* suggest a potential role of this AAA+ motor in proteome regulation and possible protein turnover, in particular of ribosomal components. This study, combining computational, biochemical, structural and microbiological approaches, will thus be of broad interest and generally appropriate for publication in *eLife*.

Essential revisions:

The reviewers felt that some of the central findings are still preliminary or based on indirect evidence, and require further support by experimental data. Before acceptance, the paper will therefore need some essential revisions, as outlined below.

1) Cpa's interaction with the 20S proteasome is shown only indirectly, through a 2-hybrid system and the inhibition of Mpa-20S mediated substrate degradation. Moreover, the functional relevance of a potential Cpa-20S complex remains to be determined.

A concern for the 2-hybrid assay is the 20S fusion construct used. The T18 subdomain of adenylate cyclase was fused to the N-termini of the 20S α subunits, even though these N-termini form the gate to the 20S core and are right at the center of the predicted binding interface for the Cpa hexamer. Such an N-terminal fusion, even when using a long linker, would be expected to interfere with or prevent proper coaxial stacking of the ATPase and peptidase. Neither this manuscript nor the previous study by Delley et al., 2014 address this issue and explain why this fusion wouldn't be a concern.

Similarly, the competitive inhibition of Mpa-20S mediated substrate degradation by large amounts of Cpa does not directly prove a functional interaction between Cpa hexamers and the 20S core particle. Analogous results would be expected if Cpa non-specifically interacted with Mpa or the substrate, and there are currently no controls presented to rule this out. For this Mpa competition assay, the authors should at least consider using the pore-2 loop mutant of Cpa to strengthen this point about Cpa hexamers docking atop the 20S core.

Importantly, the authors should aim to provide more direct evidence for a Cpa-20S interaction, either by a binding assay, by structural methods, or through functional readouts, like 20S gate opening or a potential response of Cpa's ATPase activity to 20S binding.

An additional weakness of the experimental system presented in Figure 4B is that it relies on the use of an open-gate 20S mutant. Assuming that the observed competition indeed reflects interaction of Cpa with the peptidase core, it still remains unclear whether such an interaction also occurs with wild type 20S core. This should be tested.

Of course, it would be ideal to show Cpa-dependent degradation directly, but the reviewers certainly understand that such studies are beyond the scope of this manuscript due to the lack of a model substrate and knowledge about potential cofactors, recognition motifs etc. It may be worth investigating, however, whether Cpa can be mutated like its eukaryotic and archaeal counterparts to recognize and process ssrA-tagged substrates.

2) The identification of ~ 500 proteins whose levels change in *M. smegmatis* upon Cpa knockout is a very promising finding, but it will require additional experiments to convincingly conclude that these protein levels are indeed affected by the AAA motor function of Cpa, either in a degradative Cpa-20S complex or by an isolated Cpa chaperone. In addition to the presented complete knockout, the authors should for instance consider analyzing strains carrying the pore-2 loop mutant or ATPase-deficient Cpa. Also, despite the lack of a conserved HbYX motif, it would still be worth investigating the contributions of Cpa's C-terminal tails to 20S core binding in order to get a more complete picture of this interaction. In case these tails contribute to 20S binding, mutating or truncating them would provide a very useful tool to disrupt 20S binding while maintaining normal Cpa motor function, which would allow to directly distinguish between unfolding/chaperoning versus degradative activities of Cpa in vivo.

---

## [Author Response]

Essential revisions:

The reviewers felt that some of the central findings are still preliminary or based on indirect evidence, and require further support by experimental data. Before acceptance, the paper will therefore need some essential revisions, as outlined below.1) Cpa's interaction with the 20S proteasome is shown only indirectly, through a 2-hybrid system and the inhibition of Mpa-20S mediated substrate degradation. Moreover, the functional relevance of a potential Cpa-20S complex remains to be determined.A concern for the 2-hybrid assay is the 20S fusion construct used. The T18 subdomain of adenylate cyclase was fused to the N-termini of the 20S α subunits, even though these N-termini form the gate to the 20S core and are right at the center of the predicted binding interface for the Cpa hexamer. Such an N-terminal fusion, even when using a long linker, would be expected to interfere with or prevent proper coaxial stacking of the ATPase and peptidase. Neither this manuscript nor the previous study by Delley et al., 2014 address this issue and explain why this fusion wouldn't be a concern.

We have re-cloned the fusion construct and placed the T18 domain at the C-terminus of the *prcA* gene. Otherwise the construct remained unchanged. Additionally, we included a Cpa variant with its C-terminus truncated by five amino acids in order to test whether the C-terminus is the main mediator of Cpa-CP interaction. Both the wt Cpa and the truncated variant generate an increase in β-galactosidase activity (as compared to the negative control) when co-transformed with the new prcBA fusion construct, indicating that the proteins interact in vivo (Figure 4).

Similarly, the competitive inhibition of Mpa-20S mediated substrate degradation by large amounts of Cpa does not directly prove a functional interaction between Cpa hexamers and the 20S core particle. Analogous results would be expected if Cpa non-specifically interacted with Mpa or the substrate, and there are currently no controls presented to rule this out. For this Mpa competition assay, the authors should at least consider using the pore-2 loop mutant of Cpa to strengthen this point about Cpa hexamers docking atop the 20S core.

In an attempt to address this problem, we constructed two Cpa variants: one with the D2 pore-2 loop completely removed (Barthelme and Sauer, 2013) and another variant where the loop was replaced by four amino acids (SDSG → GAGA). After expression and purification, we tested the assembly state of either variant in presence of ADP-AlF_x_, similarly as we did previously for the wild-type Cpa. Unfortunately, both of the mutants displayed an assembly defect (see Author response image 1). In this context, a diminished activity does not allow any conclusions as to the interaction specificity, as it could be the result of the assembly defect instead. We therefore did not further pursue this line of experiments and also removed the ΔD2 mutant shown previously in the two-hybrid experiment, as we feel that the reduction of interaction strength could be caused by defective assembly and the experiment is therefore inconclusive.

Importantly, the authors should aim to provide more direct evidence for a Cpa-20S interaction, either by a binding assay, by structural methods, or through functional readouts, like 20S gate opening or a potential response of Cpa's ATPase activity to 20S binding.

In order to provide a more direct evidence of complex formation between Cpa and 20S core particle, we have recorded negatively stained electron micrographs of rhodococcal Cpa (pre-assembled in presence of ADP-AlF_x_ and purified by gel filtration) in presence of rhodococcal closed-gate 20S CP (new Figure 4C). We could observe a small fraction of capped proteasomes alongside free Cpa and uncapped 20S, as shown in Figure 4C and Figure 4—figure supplement 1 (white arrows). Additionally, we used microscale thermophoresis to determine dissociation constants for the interaction of both closed- and open-gate proteasome with ^Rer^Cpa. The dissociation constants for both complexes lie in the range of 0.4 and 0.6 μM. Taken together, our data support interaction between Cpa and the proteasome that likely is of transient and very dynamic nature. The functional link will require identification of substrates or substrate classes and will have to be established in future studies. We expect that as yet unidentified adaptor complexes might be required for substrate binding.

An additional weakness of the experimental system presented in Figure 4B is that it relies on the use of an open-gate 20S mutant. Assuming that the observed competition indeed reflects interaction of Cpa with the peptidase core, it still remains unclear whether such an interaction also occurs with wild type 20S core. This should be tested.

The use of an open-gate variant in this work is unfortunately dictated by the inability of Mpa to form a stable complex with wtCP in vitro. In an effort to show that Cpa can interact with the wild-type core particle, we have incorporated the full-length CP into the new interaction experiments provided in this revision, where experimentally possible. Importantly, the new electron microscopy micrographs present interaction between Cpa and full-length, wild-type proteasomes. Additionally, the microscale thermophoresis binding curves could be obtained for both proteasome variants, showing that they exhibit similar affinity. Furthermore, the two-hybrid experiment is also performed using the closed-gate variant of the mycobacterial proteasome.

Of course, it would be ideal to show Cpa-dependent degradation directly, but the reviewers certainly understand that such studies are beyond the scope of this manuscript due to the lack of a model substrate and knowledge about potential cofactors, recognition motifs etc. It may be worth investigating, however, whether Cpa can be mutated like its eukaryotic and archaeal counterparts to recognize and process ssrA-tagged substrates.

To address this issue, we have cloned and expressed a Cpa mutant carrying a D1 pore-1 loop mutation that was previously shown to enable the eukaryotic Cdc48 to recognize ssrA-tagged substrates (TEAN → KYYG). Additionally, we removed the N-domain since its removal has been previously shown to enhance degradation activity. After confirming assembly behaviour similar to wild type Cpa (not shown), we carried out a degradation test using ssrA-tagged Mdh – the results are shown in Author response image 2. No degradation was observed.

**Author response image 2. respfig2:** 

2) The identification of ~ 500 proteins whose levels change in M. smegmatis upon Cpa knockout is a very promising finding, but it will require additional experiments to convincingly conclude that these protein levels are indeed affected by the AAA motor function of Cpa, either in a degradative Cpa-20S complex or by an isolated Cpa chaperone. In addition to the presented complete knockout, the authors should for instance consider analyzing strains carrying the pore-2 loop mutant or ATPase-deficient Cpa.

In an effort to provide evidence that many of the observed proteome changes are influenced directly by Cpa, we performed a co-immunoprecipitation experiment using Msm cells grown under the same starvation conditions as was done for the LFQ-MS. We hoped to identify Cpa binders in the wild-type cells that Cpa could be recognizing/processing at the time of starvation and the levels of which at the same time were detected as changed in the proteomics-wide approach. This would at least support direct physical interaction between Cpa and those proteins also identified in the co-immunoprecipitation. We performed the experiment in triplicate and took into consideration only those hits that occurred in at least two repetitions but not in the control co-IP (performed on the Msm Δcpa lysate). Interestingly, the results of these co-IP experiments are in line with the observation that ribosomal proteins accumulated in the *cpa*-deletion strain during carbon starvation. We observed that the co-IP elutions were also statistically enriched in ribosomal proteins (confirmed by both, STRING and DAVID tools). This would lend more support to our hypothesis that Cpa is involved in remodelling/disassembly of the ribosome under nutrient limitation.

Whether Cpa is involved as a degradative Cpa-20S complex or as an isolated chaperone or maybe a combination of both will have to be determined in a future study.

*Also, despite the lack of a conserved HbYX motif, it would still be worth investigating the contributions of Cpa's C-terminal tails to 20S core binding in order to get a more complete picture of this interaction. In case these tails contribute to 20S binding, mutating or truncating them would provide a very useful tool to disrupt 20S binding while maintaining normal Cpa motor function, which would allow to directly distinguish between unfolding/chaperoning versus degradative activities of Cpa* in vivo.

As already pointed out in our first response, we tested a CpaΔC5 truncation using the bacterial two-hybrid approach – we could not detect a significant difference between this mutant and the full-length Cpa. We do agree, however, that it would be of great interest to investigate in future work the exact binding interface between Cpa and 20S CP in order to gain a better understanding of this transient/dynamic interaction.